# Mouse models of pediatric high-grade gliomas with MYCN amplification reveal intratumoral heterogeneity and lineage signatures

Melanie Schoof [1,2], Shweta Godbole [3], Thomas K. Albert [4], Matthias Dottermusch [3,5], Carolin Walter[6], Annika Ballast[4], Nan Qin[7,8,9,10], Marlena Baca Olivera[7,8,9,10], Carolin Göbel [1,2], Sina Neyazi [1,2], Dörthe Holdhof [1,2], Catena Kresbach[1,2,5,11], Levke-Sophie Peter [1,2], Gefion Dorothea Epplen[1,2], Vanessa Thaden[1,2], Michael Spohn[1], Mirjam Blattner-Johnson[12,13], Franziska Modemann [11,14], Martin Mynarek [2,11], Stefan Rutkowski[2], Martin Sill[12,15], Julian Varghese [6], Ann-Kristin Afflerbach[1,2], Alicia Eckhardt[1,2,16], Daniel Münter [4], Archana Verma[4], Nina Struve[11,16], David T. W. Jones [12,13], Marc Remke [7,8,9,10], Julia E. Neumann [3,5], Kornelius Kerl [4,17] & Ulrich Schüller [1,2,5,17] ✉

Pediatric high-grade gliomas of the subclass MYCN (HGG-MYCN) are highly aggressive tumors frequently carrying *MYCN* amplifications, *TP53* mutations, or both alterations. Due to their rarity, such tumors have only recently been identified as a distinct entity, and biological as well as clinical characteristics have not been addressed specifically. To gain insights into tumorigenesis and molecular profiles of these tumors, and to ultimately suggest alternative treatment options, we generated a genetically engineered mouse model by breeding *hGFAP-cre::Trp53^{Fl/Fl}::lsl-MYCN* mice. All mice developed aggressive forebrain tumors early in their lifetime that mimic human HGG-MYCN regarding histology, DNA methylation, and gene expression. Single-cell RNA sequencing revealed a high intratumoral heterogeneity with neuronal and oligodendroglial lineage signatures. High-throughput drug screening using both mouse and human tumor cells finally indicated high efficacy of Doxorubicin, Irinotecan, and Etoposide as possible therapy options that children with HGG-MYCN might benefit from.

Recently, tumors originally diagnosed as primitive neuroectodermal tumors (PNETs) have been reclassified into multiple different pediatric brain tumor classes[1]. One clearly distinguishable class of these tumors are "pediatric high-grade gliomas with *MYCN* amplification", hereafter called "HGG-MYCN". This recently described tumor entity was further characterized by Korshunov et al. and Tauzière-Espariat et al. as being

highly aggressive pediatric gliomas with a poor prognosis and a median overall survival of only 14 months[2–4]. When compared to other pediatric glioma, HGG-MYCN present with the worst survival of subtypes[3]. Unlike other glioma, HGG-MYCN histologically present with undifferentiated, densely packed cell nuclei and highly circumscribed tumors without typical glial features. Molecularly, these tumors often

carry amplification of the *MYCN* gene and somatic or constitutional mutations in *TP53*[2,4,5]. Usually, these tumors are treated according to protocols for high-grade glioma with no specificities or targeted therapies available.

MYCN is a transcription factor of the MYC family, which consists of three paralogues: cellular (c-), lung-carcinoma derived (l-), and neuroblastoma-derived (n-)–myelocytomatosis (myc). Even though all three share a common structure containing so-called MYC boxes and a basic helix-loop-helix domain, they show a distinct expression profile in normal tissue and disease. MYCN is expressed in neural tissues and is essential for normal central nervous system (CNS) development[6,7]. It was discovered in 1983 as an amplified gene in neuroblastoma, a childhood tumor of the neural crest, with similarities to the *c-MYC* gene[8,9]. Shortly after its discovery, MYCN was associated with high-risk neuroblastoma and poor prognosis[10–12].

*MYCN* has also been described to be involved in a number of further tumor entities, spanning hematologic malignancies[13,14], lung cancers[15], and malignancies of the nervous system[16]. In CNS tumors, MYCN is mainly involved in tumorigenesis of retinoblastoma[17], medulloblastoma[18–22], and glioblastoma[23–25].

Although a few mouse tumor models driven by alterations of MYCN have been described, none of them have developed gliomas with similarities to human HGG-MYCN. The most prominent MYCN-driven tumor models mimic human neuroblastoma, in which MYCN expression, driven by tyrosine-hydroxylase (TH)- or dopamine-β-hydroxylase- (Dbh)-cre, leads to the development of aggressive tumors, modeling the human disease[26,27]. Forcing MYCN expression in the hindbrain by the *Glt1*-promoter induced the development of tumors, which resemble human medulloblastoma (MB) with the highest similarity to group 3 MB[28]. This mouse model is well established as a group 3 MB model although MYCN is usually associated with group 4 MB[29]. Furthermore, mutated MYCN is able to transform different types of neural stem cells (NSCs) leading to medulloblastoma- and glioblastoma-like tumors in hind- and forebrain, respectively[30]. On the other hand, forced expression of wild-type MYCN in the hindbrain or entire CNS does not necessarily lead to brain tumor development[31,32]. Given the lack of existing mouse models for HGG-MYCN on one hand and the urgent need for alternative treatment modalities for this aggressive disease on the other hand, we here aimed to develop a murine model for HGG-MYCN mimicking their human counterparts. As HGG-MYCN often carry both alterations, *MYCN* amplifications and *TP53* mutations, we generated a mouse model with combined expression of wild-type human *MYCN* and a loss of *Trp53*, which, on its own, induces glioma development in mice with a long latency and reduced penetrance[33].

In this work, we successfully generate a mouse model for HGG-MYCN by inducing *MYCN* expression and simultaneous *Trp53* deletion in *hGFAP-cre* expressing cells (*hGFAP-cre::Trp53^{Fl/Fl}::lsl-MYCN*), and show that these mice develop large forebrain tumors with 100% penetrance within 90 days. These tumors recapitulate human HGG-MYCN histologically and molecularly. We use a multi-omic approach to dissect the tumor biology of murine gliomas as well as high-throughput drug screening to identify alternative treatment options for these aggressive tumors.

## Results

### HGG-MYCN tumors represent a distinct tumor entity and frequently carry alterations in TP53 and MYCN

Studies published on the recently discovered rare HGG-MYCN entity describe a group of aggressive pediatric gliomas, which form a distinct cluster in global DNA methylation analysis[1,2,34,35]. We collected respective data from published and five in-house cases of HGG-MYCN diagnosed by DNA methylation profiling and used a reference set of pediatric brain tumors to confirm the distinctiveness of these tumors by Uniform Manifold Approximation and Projection (UMAP). We

included the most common brain tumors as well as potential differential diagnoses and analyzed the global DNA methylation of 2514 tumors, including 47 HGG-MYCN (Fig. 1a). Clinical information of the HGG-MYCN patients revealed a median age of 8 years with only 4 out of 89 patients older than 20 years (range: 1–56 years, Fig. 1b). Sex distribution was almost balanced (Fig. 1c). DNA sequencing information was available for 47 cases, and we found *TP53* mutations in 68% and *MYCN* amplifications in 60% of those cases (Fig. 1d, e). Thirty-six percent of the analyzed cases carried both a loss of *TP53* and an amplification of *MYCN* (Fig. 1f). HGG-MYCN were detected throughout the entire brain with a preponderance of tumors occurring in the frontal and temporal lobe (Fig. 1g). The tumors presented with multiple chromosomal aberrations and, in part, a clearly visible *MYCN* amplification ($n = 19$) as visualized in a heatmap of the copy number variations of 47 tumors as well as by Fluorescence-in situ hybridization (FISH) or immunohistochemistry (IHC). IHC showing accumulated p53 indicating a mutated *TP53* (Fig. 1h–l).

### hGFAP-cre driven loss of p53 and expression of MYCN induce brain tumor formation in mice

In order to generate an appropriate mouse model for HGG-MYCN, we exploited the Cre-LoxP system to simultaneously induce the deletion of *Trp53* and force expression of human wild-type MYCN under the control of the *hGFAP* promoter. This promoter is active from embryonic day (E) 13.5 onwards and targets radial glia, which later differentiates to distinct cell types of the CNS, including neurons, oligodendrocytes, astrocytes, and adult neural stem cells[36]. Loss of p53 is accomplished by an allele carrying loxP-sites after exon 1 and 10 in the *Trp53* gene, leading to a large deletion in the gene upon recombination. The expression of MYCN is achieved by inserting an allele into the ubiquitously expressed *Rosa26* locus, where a stop codon flanked by two loxP sites is preceding the *MYCN* open-reading-frame (Fig. 2a).

All *hGFAP-cre::Trp53^{Fl/Fl}::lsl-MYCN* animals ($n = 24$) developed neurological symptoms within 90 days of life with the first animals being symptomatic around postnatal day 40 (Fig. 2b). The animals abruptly presented with hydrocephalus and akinesia and were sacrificed as soon as symptoms appeared. Upon necropsy, large forebrain tumors and enlarged ventricles were macroscopically found (Fig. 2c). Midbrain and hindbrain as well as the spinal cord and other organs appeared macro- and microscopically normal. This was of particular interest, since *MYCN* amplifications and/or *TP53* mutations have also been observed in a subset of spinal ependymoma as well as in pediatric medulloblastoma[21,22,37]. In this model, none of the microscopically analyzed mice ($n = 17$) showed overexpression of MYCN in these regions.

Histological examination of the murine brains revealed large, cell-dense tumors of the forebrain with a heterogeneous cell morphology (Fig. 2d, S1). To determine the timing of tumor initiation, we sacrificed animals at postnatal day (P) 7 and 37 prior to the establishment of any symptoms. At P7, no tumor lesion was detected by histological examination of the brains of six mice, whereas at P37, tumor lesions were detected in all analyzed mice ($n = 5$). These small tumor lesions presented with densely packed, highly proliferating cells in the outermost layer of the olfactory bulb (OB) (Fig. S1).

Genetic analysis of the tumors developing in *hGFAP-cre::Trp53^{Fl/Fl}::lsl-MYCN* mice showed a recombined stop codon as well as a recombined *Trp53* allele as expected (Fig. 2e). We also analyzed potential copy number variations (CNVs) of three representative mouse tumors (Fig. 2f). Due to the genetic engineering, all three mouse tumors revealed an amplification of the *Rosa26*-locus with the inserted *MYCN* on chromosome 6 (asterisk in Fig. 2f). Moreover, they showed chromosomal aberration private to one tumor as well as recurrent chromosomal aberrations appearing in at least two of three tumors at chromosomes 7, 14, or 16 (Fig. 2f). This suggests that the combination of *MYCN* and *Trp53*

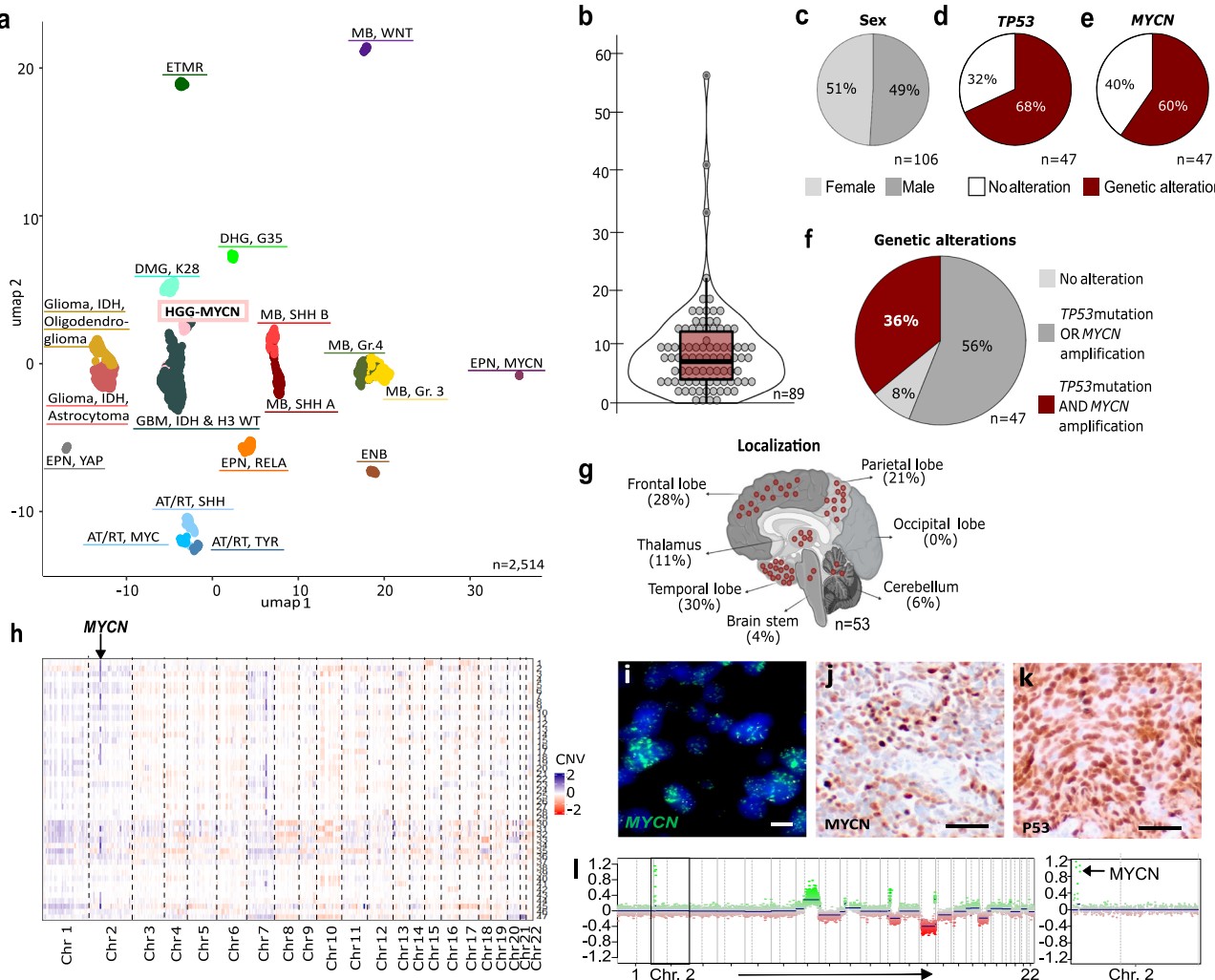

**Fig. 1 | Tumors of the DNA methylation class HGG-MYCN carry MYCN amplifications and TP53 mutations. a** UMAP of global DNA methylation profiling of 2514 cases of multiple brain tumor entities including the most common brain tumors as well as potential differential diagnoses including 47 HGG-MYCN using the 10,000 most differentially methylated CpG sites. **b** Age distribution of *n* = 89 HGG-MYCN tumors with a median age of 8 years (bounds of box = 3–13, whiskers = min:1, max:22). **c** Sex distribution of 106 HGG-MYCN. **d** 47 HGG-MYCN were screened for TP53 mutations, of which 68% carried a mutation. **e** The same 47 cases as in (**d**) were analyzed for their MYCN status. **f** Of the same 47 cases shown in (**d**) and (**e**), only 8% carried no TP53 or MYCN alteration, whereas 36% of those carry both alterations. **g** Tumors can be found throughout the entire brain with the majority of cases located in the temporal and frontal lobes. **h** Heatmap showing copy number variations of 47 HGG-MYCN. The copy number profile of such tumors is imbalanced with a clearly visible *MYCN* amplification (Chr. 2), highlighted with the arrow. *MYCN* amplifications can also be detected by FISH analysis (representative case shown in (**i**), three independent tumors showing this amplification were analyzed), while IHC may serve as a surrogate marker (representative case in (**j**), the same three tumors were analyzed). **k** Nuclear p53 accumulation indicating impaired p53 function can be detected by IHC (representative case, three independent tumors were analyzed). **l** Representative CNV plot of a HGG-MYCN with a magnification of chromosome 2 with the *MYCN* amplification (CNV plots of 47 tumors were generated). Scale bar in **i** = 5 μm, in **j** & **k** = 50 μm. Source data are provided as a Source Data file.

alterations induce further genomic changes that may be needed for tumor development.

We next characterized the mouse tumors by microscopy to see whether they resemble their human counterparts. Tumors of both species showed irregularly shaped, densely packed cell nuclei (Fig. 2g, n) and were positive for MYCN (Fig. 2h, o). They were also highly proliferative (Fig. 2i, p) and expressed Nestin (Fig. 2j, q), SOX2 (Fig. 2k, r), OLIG2 (Fig. 2l, s), and GFAP (Fig. 2m, t). Other histological markers were also similarly expressed in mouse as well as human tumors: Neurofilament, NeuN, and Synaptophysin were not expressed but tumors of both, mouse and human, expressed TubB3 (Fig. S2).

So far, the cell-of-origin as well as time and place of tumor onset for HGG-MYCN is unknown. Therefore, we decided to investigate tumor development in other cell populations of the developing CNS by breeding animals with the same genetic alterations in other target cell populations. We bred *Sox2-cre::Trp53^FI/FI^::lsl-MYCN* and *Blbp-cre::Trp53^FI/FI^::lsl-MYCN* mice, which initiate recombination upon E6.5 or E9.5, respectively. The *Sox2*-mediated recombination was embryonically lethal (Fig. S3a) with severely underdeveloped animals at E16.5 (Fig. S3b–d). The *Blbp*-mediated recombination also led to prenatal lethality except for two animals surviving until P0 and two until P18, on which they presented with hydrocephalus (Fig. S3e). Histologic examination of the P18 brains showed no signs of tumor development, suggesting rather a developmental defect as the cause for the hydrocephalus (Fig. S3f–h).

## Mouse HGG-MYCN molecularly resemble their human counterparts

We next investigated whether murine and human tumors shared similarities regarding their DNA methylation profiles. We used bead chip arrays to detect the methylation status of 285,000 CpG sites of

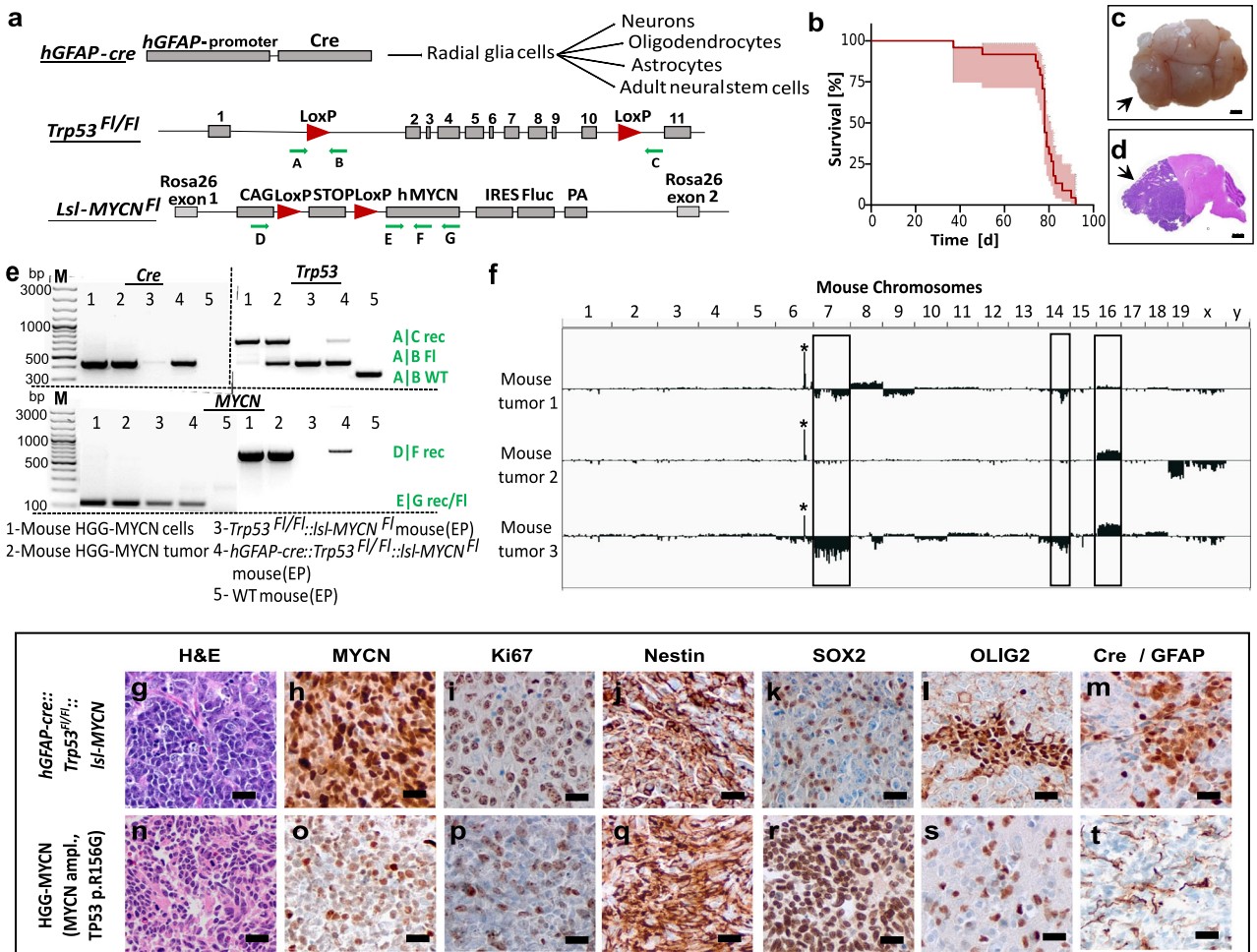

**Fig. 2 | Mouse HGG-MYCN develop within the first 100 days of life and match their human counterparts histologically. a** Genetics of the HGG-MYCN mouse model. Cre is expressed under control of the *hGFAP* promoter, which targets different cell populations shown on the right. The floxed alleles of *Trp53* and *MYCN* are depicted, loxP sites are shown as red arrowheads, and primers for genotyping and proof of recombination are shown as green arrows. **b** Kaplan–Meier survival curve of mice with HGG-MYCN ($n = 24$) as percent survival, light red area showing the asymmetrical 95% confidence interval. **c** Macroscopic image of a mouse brain carrying an HGG-MYCN (arrow). **d** Hematoxylin and Eosin (H&E) stained brain with a large forebrain tumor (arrow). **e** Representative PCR result of genotyping and the detection of allele recombination. Results are shown for cultured mouse tumor cells (1), fresh mouse tumor tissue (2), ear biopsy of a mouse not carrying the Cre recombinase (3), ear biopsy of a mouse with an HGG-MYCN (4), and (5) a wild-type

mouse carrying none of the transgenes. Bands result from the primers indicated by green arrows in (**a**). The same PCR was performed for all animals generated in the study including the $n = 24$ animals included in the Kaplan-Meier survival analysis. **f** Copy number variation plots of three mouse HGG-MYCN. An amplification of the *Rosa26*-locus, in which the *MYCN* is inserted, is visible in all three samples (marked by the star). Other recurrent copy number changes are observed in at least two of the tumors (marked by the rectangle). **g–t** Immunohistochemical comparability of mouse and a human HGG-MYCN. The pictures show representative micrographs, all stainings were performed independently on at least three samples. EP ear punch, Fl Floxed allele, M marker, Rec recombined allele, WT wild-type allele. The scale bar in c and d corresponds to 2 mm, and the scale bar in (**g–t**) corresponds to 20 µm. Source data are provided as a Source Data file.

the mouse genome. Of these sites, 141 CpG sites were identical to the human 850k (EPIC) array. The beta-values of these 141 sites were sufficient to distinguish different human brain tumor entities according to the visualization via UMAP. When clustering the methylation data of the murine tumors with the most common pediatric brain tumors as well as potential differential diagnosis, mouse HGG-MYCN (hot pink) showed the highest similarity to the human HGG-MYCN (light pink, Fig. 3a). As a control, we included tumors of *Math1-cre::SmoM2*[Fl/wt] mice, which are a well-known murine model for sonic hedgehog medulloblastoma (SHH-MB)[38] and which were located nearest to human SHH MB, as expected (Fig. 3a).

Next, we generated transcriptomic profiles of the murine tumors and compared them to gene expression data of human HGG-MYCN and other pediatric brain tumors[1]. The visualization of gene expression profiles by UMAP analyses revealed the highest similarity between mouse HGG-MYCN (hot pink) to their human

counterparts (light pink, Fig. 3b). Again, SHH-MBs served as an internal control with high similarities between murine (red) and human SHH-MB tumors (dark red). We further analyzed the similarity of gene expression profiles by calculating the Euclidean distance of averaged transcriptome data per group and performed an agreement of differential expression (AGDEX)[39] analysis. Both analyses confirmed the high similarity between mouse and human HGG-MYCN (Fig. 3c, d, Euclidean distance: HGG-MYCN – mouse HGG-MYCN = 24.4, AGDEX: HGG-MYCN – mouse HGG-MYCN, AGDEX cos: 0.190659811, *p*-value = 0.007). When we compared *MYCN* expression in human and mouse HGG-MYCN relative to other human gliomas or healthy mouse tissue of the OB and cerebellum and the SHH-MB mouse model, and performed gene set enrichment analysis (GSEA) for MYCN target genes, we found that MYCN itself as well as its target gene sets are significantly enriched in tumors of both species, human and mouse (Fig. 3e, f, two-sided Welch's-*t*-test,

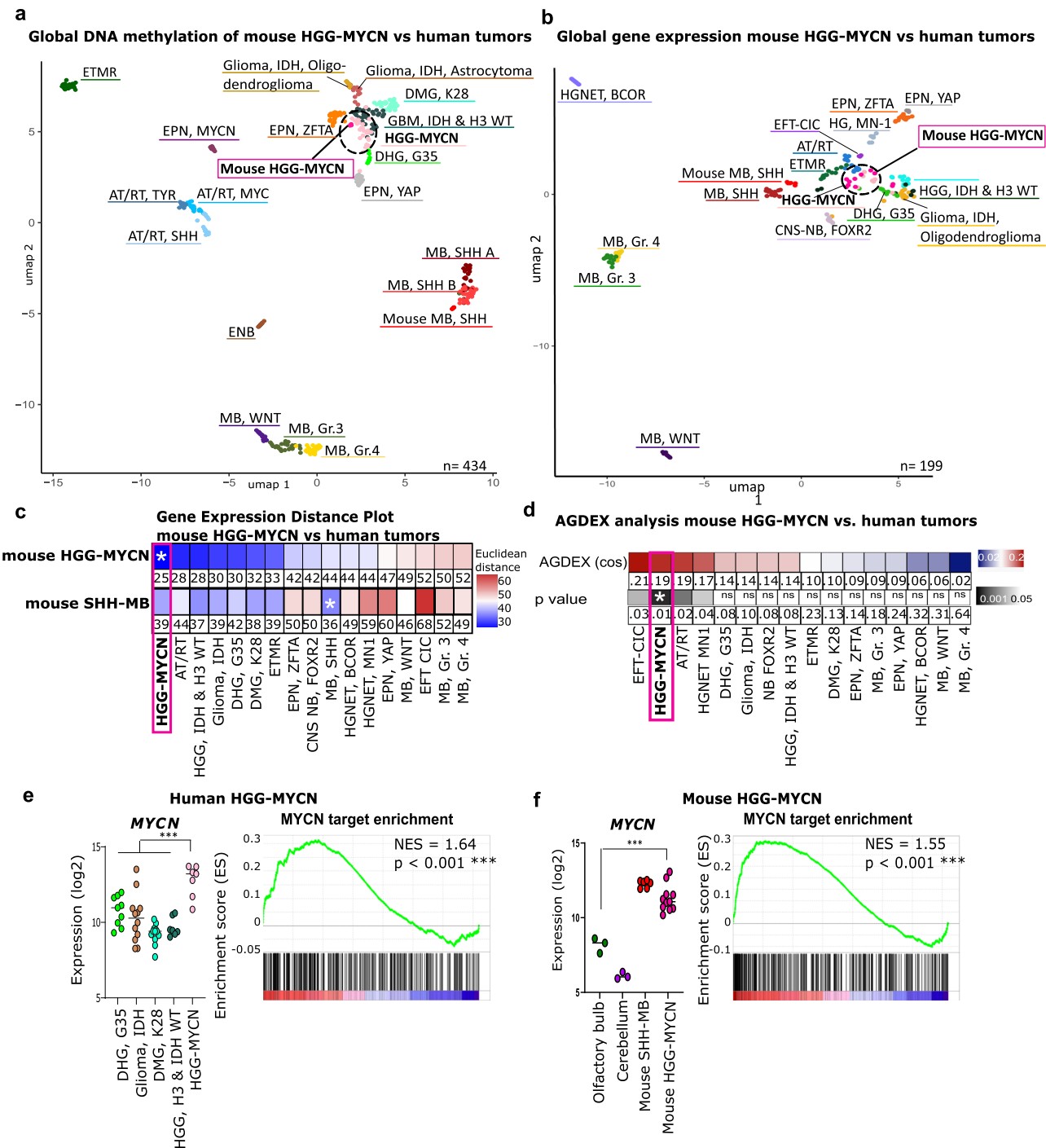

**Fig. 3 | Mouse HGG-MYCN match their human counterparts molecularly.**
**a** UMAP of global DNA methylation. Betavalues of 141 identical CpG sites between mouse and human were used for comparison of similarity. Mouse methylome data were generated with the Illumina Mouse Methylation bead chip array and compared to human data generated with the EPIC array. The three mouse tumors (in hot pink) show most similarity to the human HGG-MYCN group (pink). **b** UMAP of global gene expression data. Eleven mouse tumors were profiled by RNA sequencing and their gene expression profile was compared to published gene expression data of human „CNS PNET" tumors 1. Data were normalized for interspecies differences by employing RNA Seq. data of *Math1-cre::SmoM2^{Fl/wt}* mouse tumors and human SHH medulloblastoma. **c** Distance plot of mouse HGG-MYCN and mouse SHH-MBs and human tumors based on the 500 most significantly expressed genes. Mouse tumors showed most similarity to human HGG-MYCN and human SHH-MB,

respectively. The values of the Euclidean distance are displayed, and the asterisk marks the smallest values and thereby highest similarity. **d** AGDEX analysis of mouse HGG-MYCN also shows the high similarity between murine and human tumors (analysis based on 14,416 orthologous genes). The values of the AGDEX analysis with their respective *p*-values are given. The asterisk marks the smallest *p*-value and thereby highest similarity. **e**, **f** Human and mouse HGG-MYCN show significantly higher *MYCN* expression compared to other glioma entities or control tissue (from mouse olfactory bulb (OB) or cerebellum) and comparable expression to the SHH-MB mouse model as well as highly significant enrichment for MYCN target genes. Human tumors: non-mycn, $n = 40$, HGG-MYCN $n = 7$. Mouse data: OB tissue $n = 3$, mouse HGG-MYCN $n = 11$., two-sided Welch's-*t*-test, human: $p = 0.0002$, 95% confidence interval = 2.001 to 4.320, murine= $p = 0.0002$, 95% confidence interval = 2.143 to 4.085. Source data are provided as a Source Data file.

human: $p = 0.0002$, 95% confidence interval = 2.001 to 4.320, murine= $p = 0.0002$, 95% confidence interval = 2.143 to 4.085).

## Single-cell transcriptomics of mouse HGG-MYCN revealed a high intratumoral heterogeneity with neuronal and oligodendroglial tumor cell population

As knowledge about the cellular architecture and the cell-of-origin of HGG-MYCN is lacking and no single cell RNA sequencing (scRNA seq) data of HGG-MYCN is available, we employed scRNA seq to dissect the cellular composition of these murine tumors. We profiled the single cell transcriptomes of seven mouse tumors using 10X genomics technology. The cells were harvested from OB/tumor tissue on P43 ($n = 2$), P70 ($n = 2$), P77 ($n = 2$), and P92 ($n = 1$). ScRNA seq. resulted in 23 distinct clusters (Fig. 4a). The tumor cell clusters were clearly identified based on their increased expression of human *MYCN* and the firefly luciferase (*FLUC*) reporter gene (Fig. 4b). The remaining clusters were unequivocally assigned as non-malignant cells of different types by a combination of differential gene expression (DGE) analysis and comparison with known highly specific marker genes of distinct tumor microenvironment (TME) cell types (Fig. 4c, S4).

Based on their similarities in the gene expression patterns, the seurat algorithm divided the tumor cells into distinct clusters (Fig. 4a–c). These tumor cell clusters were split into three larger areas: clusters 0 and 13 were composed only of cells of the most mature tumor (P92 tumor), clusters 1/2/3/4/9/15/18 were composed of cells resembling oligodendrocytic-lineage (OL) cells and cluster 5 expresses a neuronal-like signature.

In order to define the gene expression signatures of the different tumor cell populations, we employed a combination of DGE and functional gene network analysis as well as gene ontology (GO) annotation to identify the differences and similarities of distinct clusters within the murine HGG-MYCN (Fig. 4d). These analyses clearly show an oligodendrocytic tumor cell population, a neuronal tumor cell population, and a third tumor population, which is transcriptionally clearly distinct and expresses markers of neither cell lineage. Tumor cells of this cluster were solely harvested from the most mature tumor of a 92-day-old mouse.

In order to determine the potential cellular origin of the murine tumor cells, we compared gene expression profiles of the different tumor cell clusters to a reference dataset of the ventricular zone/ subventricular zone (VZ/SVZ) of the mouse. We used logistic regression to identify similarities between the reference clusters and our tumor cells. This analysis reveals a high similarity of HGG-MYCN tumor cells with adult neural stem (aNSC) and transit-amplifying neural precursor cells (TAC). Some tumor cell clusters (5,0,13,3) also show similarities to neuroblasts (NB) and some clusters to cells of the oligodendrocytic lineage (1,15,0,13,3). These results suggest that murine HGG-MYCN may originate from multipotent stem or progenitor cells in the SVZ that differentiate into both glial and neuronal lineages.

To understand the tumor development further, we investigated the tumor cells of the different stages of tumor development. The comparison of the TME and the tumor cell clusters at P43, P70, P77, and P92 revealed a reduction of TME cells throughout tumor development and a change in tumor cell populations (Fig. 4f). The tumor cells of the youngest mice are mostly OL-like. At P70, the NB-like tumor cell population appears which is also detectable at P77. At P92, only a small proportion of tumor cells can be assigned to the OL- or NB-like type but instead most cells belong to a unique cell population which we therefore called P92-tumor.

## Sensitivity of HGG-MYCN cells to Doxorubicin, Etoposide and Irinotecan in an in vitro drug screen

Next, we used our mouse model to identify improved therapeutic options to treat HGG-MYCN, since standard treatment is still inefficient. For this, we performed a high-throughput drug screening of 639 compounds in a human HGG-MYCN cell line (pbt-04) and a cell line isolated from our mouse model (pn003, Fig. 5a, Supplementary data 1). To expedite clinical use, we analyzed drug response using a clinical anticancer library comprising nearly 80% of FDA-approved drugs. The chosen drugs encompass both standard chemotherapy and targeted therapy for various types of cancer. The selection doesn't limit to the blood-brain barrier permeability since, due to the recent development of materials science and nanotechnology, multiple strategies could be used to deliver drugs across BBB.

We determined the drug response by comparing the normalized area under the curve (AUC) of a cell viability assay. We then focused on the 100 compounds, that were most efficient in reducing cell viability in both, mouse and human cells. Among these 100 compounds, we found 18 drug classes represented by more than two drugs, suggesting a specific mechanism of action against these tumors (Fig. 5b). Thirty of these drugs are FDA approved, and for 14 of those, CNS permeability has been described (Fig. 5c). By focusing on the FDA approved drugs, we identified 12 compounds that are already used for the treatment of pediatric brain tumor patients and can therefore be considered relatively safe to use also in HGG-MYCN patients (Fig. 5d, e).

The top three compounds, that are FDA approved and used to treat pediatric brain tumor patients, are Doxorubicin, Etoposide, and SN38 as the active metabolite of Irinotecan. All three compounds showed high efficacy against our HGG-MYCN cell lines, mixed response in other adult and pediatric glioblastoma reference cells and had almost no effect on healthy fibroblasts (Fig. 5f–h).

## Discussion

We show that combined loss of p53 and forced expression of MYCN in neural precursor cells is sufficient to drive brain tumor formation. Mice carrying these alterations in cells targeted by *hGFAP*-driven recombination develop large forebrain tumors. These murine tumors show a similar histology and marker expression as human tumors of the methylation class "pediatric high-grade glioma MYCN". In addition, we found the highest similarity between mouse tumors and HGG-MYCN compared to other aggressive pediatric brain tumors in DNA methylation and gene expression profiles.

We investigated the single cell transcriptomic landscape of the tumors and could show a tumor evolution of the mouse tumors by analyzing different mouse ages and stages of tumor development. We included mouse samples of early tumor onset at P43, further developed tumors at P70 and tumors of symptomatic mice at P77 and P92. During tumor development, we observed changes in the TME but also a change in the tumor cell populations. At P92, a unique tumor cell population is detected. This mouse was exceptional in survival, as only 4.4% of animals survive until P90. This might explain the differentness of the observed tumor cell population from the other six mouse tumors but as we could only generate data of this one exceptionally long surviving mouse, we cannot draw definite conclusions.

The age of disease onset differed between mice and humans, with children having a median age of disease onset of eight years and mice becoming symptomatic in adolescence. This difference might be due to species differences in maturation of neural cells or depending on environmental influences. As the age of disease onset is before adulthood in both species and the molecular biology of mouse and human tumors is very similar, we consider our model as a reliable model for the human disease.

Alterations in *MYCN* and *TP53* also occur in other human brain tumor entities, such as medulloblastoma. To investigate whether the same transgenes will lead to different brain tumors in other target cells and further characterize the cell-of-origin of our tumors, we used different promoters to drive Cre expression in the developing brain. We employed the *Sox2* promoter, which is expressed earlier in embryonic development (E6.5 instead of E13.5), and the *Blbp* promoter, which is expressed from E9.5 onwards. In fact, Swartling et al. described that

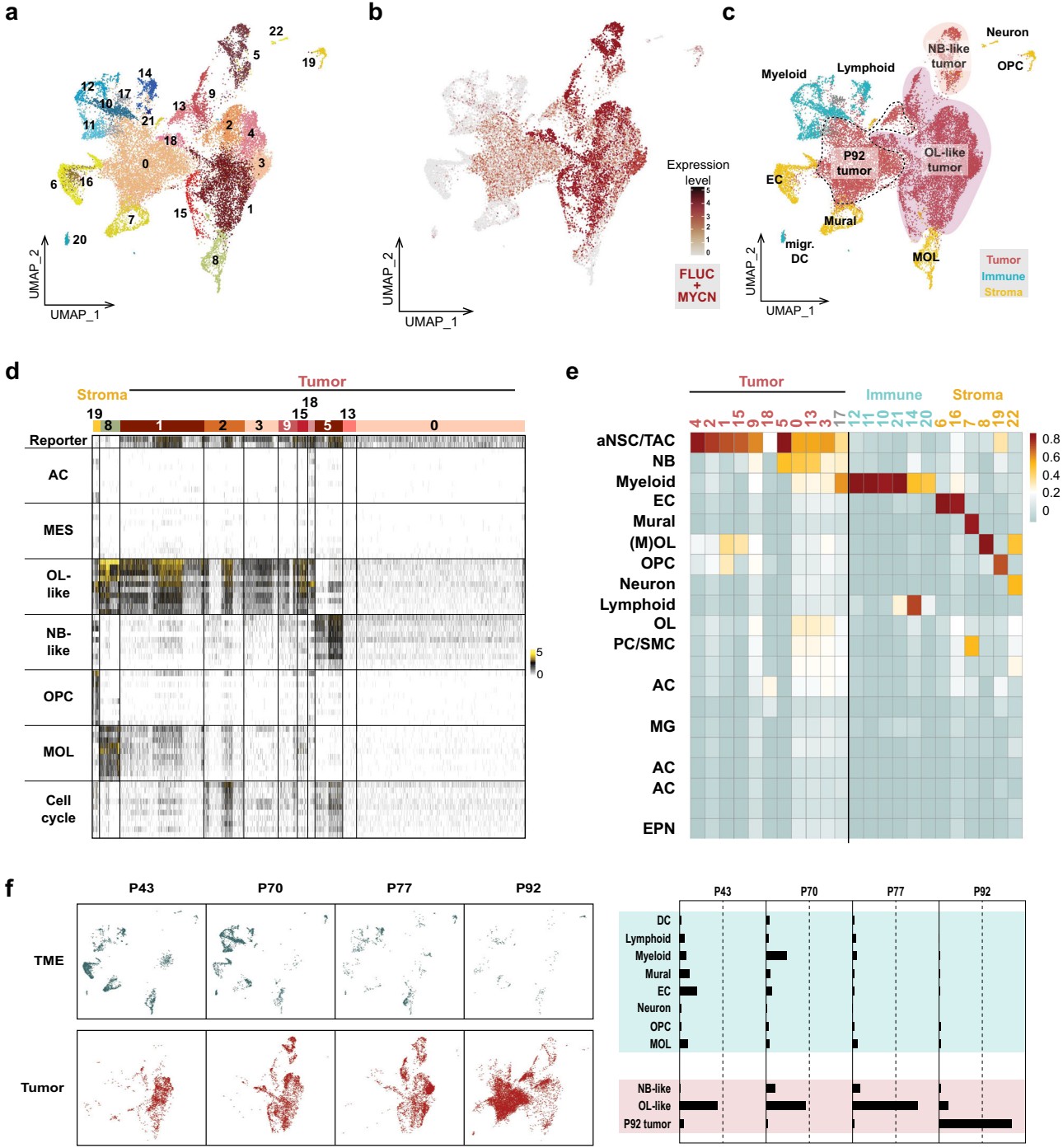

**Fig. 4 | Mouse HGG-MYCN reveal a high intratumoral heterogeneity with oligodendroglial and neural cell populations and a time-resolved change in tumor composition. a** UMAP of single cell RNA sequencing data of seven mouse HGG-MYCN (2xP43, 2xP70, 2xP77 and 1x P92) including 24.938 cells. 23 cell clustered were identified by the Seurat algorithm. **b** To identify tumor cell clusters, expression of human *MYCN* and Luciferase (*FLUC*), were plotted. **c, d** Cell clusters were annotated by analysis of marker gene expression. This revealed immune as well as stromal cell clusters and three main superclusters of tumor cells. Tumor superclusters consist of an oligodendroglial-like, a neuronal-like, and a cluster which was only detected in the most mature mouse tumor. **e** Cell clusters were compared to a reference atlas of the VZ/SVZ of the mouse by logistic regression. Similarity in gene expression is displayed in red, less similarity in blue. Mouse tumor

cell clusters show similarity to precursor cells of the stem cell niche, suggesting a tumor origin in this region. **f** UMAP and bargraph depicting the changes in cell composition of samples of different mouse ages. The TME content is reduced in later tumor stages. Early tumors are only OL-like, during tumor development, an NB-like population appears. At P92, a unique tumor cell population is detected, showing neither OL nor NB-like features. AC astrocyte, aNSC adult neural stem cell, DC dendritic cell, EC endothelial cells, EPN ependymal cells, MES mesenchymal, MG microglia, migr. DC migratory dendritc cells, MOL mature oligodendrocyte, mural fibroblasts, pericytes, smooth muscle cells etc., NB neuroblast, OL oligodendrocytic, OPC oligodendrocytic precursor cell, PC pericytes, SMC smooth muscle cells, TAC transit amplifying cell, TME tumor microenvironment, UMAP uniform manifold approximation and projection.

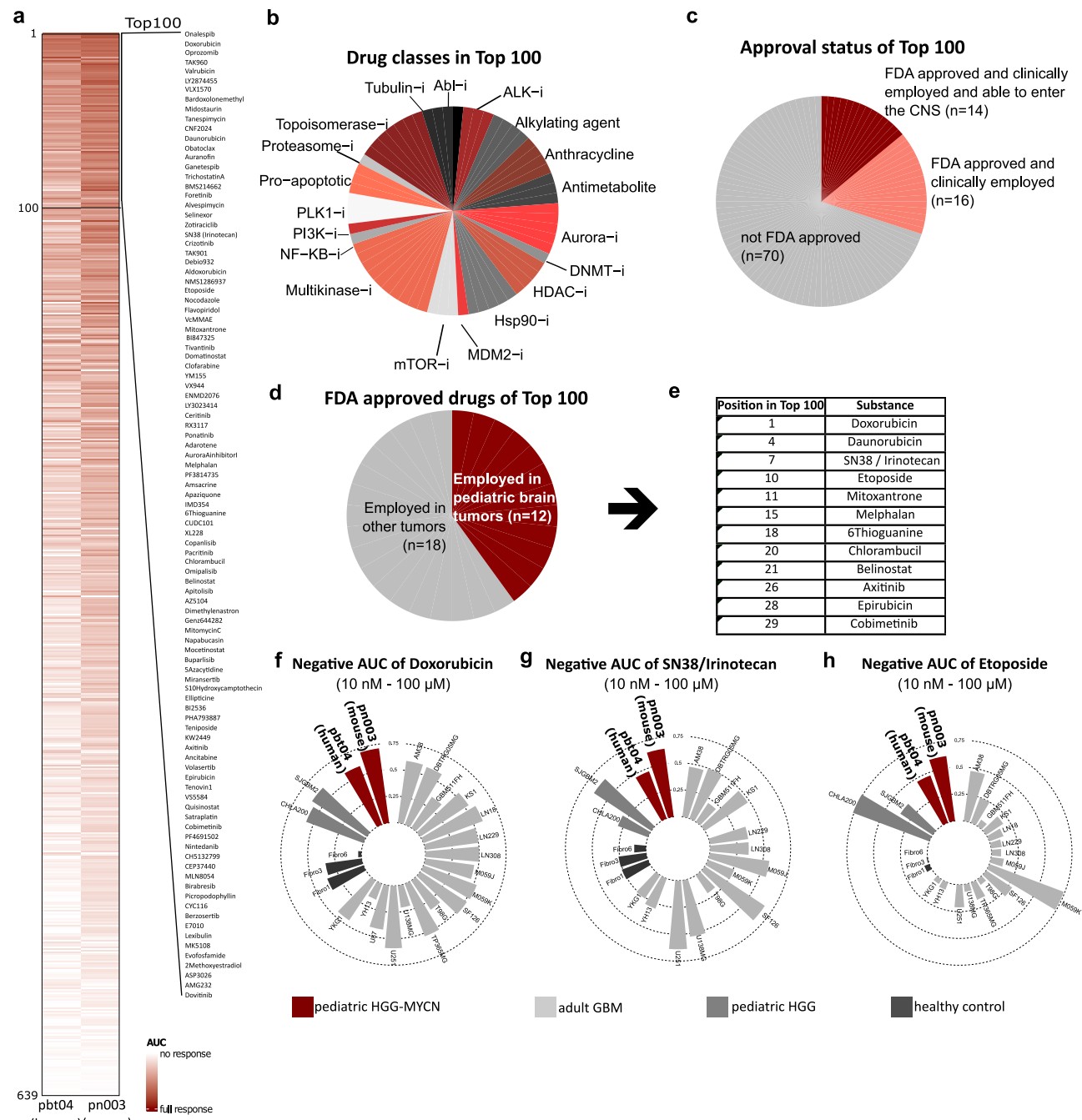

**Fig. 5 | High throughput drug screen indicates efficacy of Doxorubicin, Etoposide, and Irinotecan for the treatment of HGG-MYCN. a** Heatmap of the AUC in a drug screen of 639 substances in human (pbt-04) and murine (pn003) HGGMYCN cell lines. A darker color indicates a stronger response. **b** Among the top 100 substances of the drug screen, we detected 18 drug classes with at least two substances. The most prominent were anthracyclins, aurora inhibitors, Hsp90-inhibitors, multikinase inhibitors, and topoisomerase inhibitors. **c** Of the top 100 most effective substances in the screen, 30 were FDA approved. Of those, 14 have been described to be delivered into the CNS. **d** Of the 30 FDA-approved substances, 12 have been used to treat pediatric brain tumor patients as depicted in (**e**). **f–h** The top three substances Doxorubicin, SN-38 (active metabolite of Irinotecan) and Etoposide are efficiently impairing growth of HGG-MYCN cells, while showing a mixed effect on other glioma cell lines and almost no effect on healthy fibroblasts. Depicted is the negative AUC (1 minus the respective AUC). AUC area under the curve. AUC values are supplied in Supplementary data 1. Source data are provided as a Source Data file.

MYCN drives different brain tumor types depending on the cell of origin[30]. However, although both promoters have previously been described to drive medulloblastoma formation in mice[40,41], none of the generated animals developed any brain tumors. The embryonal lethality of MYCN expression and TP53 loss in Sox2- as well as Blbp-expressing cell populations implies that the physiologic expression of at least one of the two genes is essential during normal CNS development in this timeframe. This indicates a similar function in human

CNS development which could be a hint to the time of human HGG-MYCN development.

We particularly examined the cerebellum and the spine of our *hGFAP-cre::Trp53^{Fl/Fl}::lsl-MYCN* mice for signs of tumor development. Since no tumors were detected we conclude that loss of p53 and overexpression of MYCN, despite being expressed there, is either not sufficient for the development of medulloblastoma or spinal ependymoma or other promoters may be needed. However, we cannot rule

out that *hGFAP-cre::Trp53^Fl/Fl::lsl-MYCN* mice died from their forebrain tumors prior to the development of further lesions. Together with the scRNAseq data, we can deduce that the cell of origin for the murine HGG-MYCN is most likely a relatively undifferentiated precursor cell that is able to differentiate into the oligodendroglial and neuronal lineage as we identified cells differentiating along both lineages in our tumors.

We neither detected any glioma in the cerebellum of the mice, although in humans, at least a few cerebellar cases of HGG-MYCN have been described in the literature[42]. In our mice, tumors appear to initiate in the OB, which may be caused by differences in the sensory input of mice and humans. Brain tumor development and especially glioblastoma formation is known to be dependent on sensory input, and in humans, other brain regions, such as visual and hearing systems, might be more prone to tumor development[43,44]. In addition, tumor development in our mouse model coincides with the area of adult neurogenesis in mice, which is highest in the SVZ, from where cells migrate into the OB. In humans, the highest rate of adult neurogenesis can be found in the striatum and almost no neurons are added to the OB after initial development[45]. This species difference in neurogenesis might also contribute to differences in tumor location.

Concluding, our mouse model of HGG-MYCN has limitations in modeling the human disease. The genetic alteration driving the mouse tumors are observed in humans, but the combination of both is only found in 36% of human tumors. Therefore, our mouse model is genetically only mimicking this subset of human tumors. In addition, also the age of tumor onset, the localization of tumors, as well as the exact copy number alterations are different between mouse and human HGG-MYCN. We show similarities between the two species on a molecular level including gene expression and DNA methylation, but these methods are also not a definite proof for the translational relevance of our model. However, all employed methods for comparing mouse and human tumors including histology, transcriptomic, and epigenomics indicate a resemblance of human HGG-MYCN by mouse HGG_MYCN. The observed differences most likely depend on species-dependent differences in neurogenesis and external stimuli as discussed above. A genetically engineered mouse model has inherent limitations as mice are different from human in many aspects. Therefore, conclusions drawn from mouse models can only answer some questions of human tumor development. Anyways, modeling a tumor in an intact organism with a functional immune system and a complete organ system, will help understand aspects of the tumor. Employing a mouse model together with other techniques of modeling human tumors will help increasing the treatability and survival of patients. MYCN was discovered in 1983 by its tumor-driving role in neuroblastoma[8,9], a tumor of the sympathetic nervous system, and mouse models, in which tumors were initiated by amplification or forced expression of MYCN, were neuroblastoma models. Apart from these murine neuroblastoma, a few MYCN-driven brain tumor models have been described. One model, generated by transduction of NSCs with a mutationally stabilized MYCN and transplantation of respective cells into nude mice, generated forebrain tumors with features of glioma. However, at that time, techniques were limited to thoroughly compare the molecular landscapes of mouse and human tumors[30].

We combined *MYCN* amplification with an *hGFAP*-mediated loss of *Trp53*. The latter alone has been described to induce formation of IDHwt/H3wt glioblastoma with a penetrance of 100%, although tumor formation occurs significantly later in life than tumor development in our HGG-MYCN mice[33]. So, neither a *hGFAP*-mediated *MYCN* amplification[32] nor a *hGFAP*-mediated *TP53* loss[33] is sufficient to drive HGG-MYCN formation. Only the combination of both alterations cooperate to induce such tumors. A possible mechanism of how these two alterations co-act to induce tumor formation is that MYCN drives cell proliferation, but only in cells that are genetically unstable for example by a loss of *Trp53*. This increased proliferation can lead to

tumor formation, whereas cells with a functional p53 will activate cell cycle checkpoints and prevent uncontrolled proliferation.

We show that cells derived from our mouse model can be cultivated in vitro and be employed for preclinical testing, which is required as a preparation for the in vivo application of potential drugs.

We employed our mouse model and a human cell line to find potential therapies in a high-throughput drug screen. Thereby, we identified Etoposide, Doxorubicin, and Irinotecan as potential treatment options in patients with HGG-MYCN. All these substances are used in the treatment of pediatric brain tumors and can therefore be considered safe to use. We think that using these substances instead or in addition to the current treatment with radiation and optionally Temozolomide might substantially improve patient survival, although further in vivo studies are required. These in vivo studies could include the comparison of the current treatment with radiation and Temozolomide and Irinotecan, Etoposide, Doxorubicin and any combination of those substances. In addition, more targeted treatments described for other MYCN driven tumors like for example AURKA-, CDK2/9- and DNA replication repair inhibitors could be tested. In addition, it might be useful to employ our mouse model as well as PDX models of HGG-MYCN in such treatment studies. Our here described mouse model mimics the human disease regarding histology, DNA methylation as well as gene expression. In summary, we believe that it can serve as a reliable preclinical model for this poorly understood brain tumor entity.

## Methods

All work performed in this project complies with all relevant ethical regulations including animal protection laws of the state of Hamburg and the local ethical standards and regulations at the University Medical Center Hamburg-Eppendorf (§12 HmbKHG). Only anonymized human data were included in the project and informed consent of all human patients was obtained.

### Transgenic animals

The generation of *hGFAP-cre*[36], *Blbp-cre*[46], *Sox2-cre*[47], *Trp53^Fl/Fl*[48], and *lsl-MYCN*[26] as well as *Math1-cre::SmoM2^Fl/wt*[38] transgenic mouse lines has been described previously. All animal procedures were performed in accordance with applicable animal protection laws and approved by the state of Hamburg (Reference N2019/99). Genotyping was performed by polymerase-chain reaction with primer pairs described in the original publications (Supplementary Table 1). For all analyses, mice of both sexes were used in equal numbers. The mice were kept at 22 °C and 45–65% humidity on a 12 h light/dark cycle and water and food was available ad libitum. The animals were monitored daily by experienced personnel. All animals were sacrificed upon detectable neurological symptoms including ataxia and hydrocephalus as approved by the state of Hamburg. The termination criteria defined were never exceeded in the study.

### Human cell line culture

Pbt04 cells (SCRI, Brain tumor resource lab) were grown in NeuroCult NS-A Basal Medium (Human) supplemented with NeuroCult NS-A Proliferation Supplements−Human, Epidermal Growth Factor (EGF); Fibroblast Growth Factor (FGF) and penicillin/streptomycin (P/S) in laminin coated flasks.

### Primary murine tumor cell culture

For the establishment of murine tumor cell lines, fresh tumors were dissected and dissociated with Accutase and DNase. After dissociation, cells were seeded in mouse NSC medium (DMEM/F12 + Glutamax, HEPES, MEM-Non Essential Amino Acids, P/S, B27 Supplement, EGF, FGF) and grown in a humidified incubator at 37 °C. As soon as large spheres formed, they were dissociated with Accutase and re-seeded.

## High-throughput drug screening

High-throughput drug screening using an in-house semi-automated platform was performed. For this approach, we measured the drug responses of pbt-04 and a murine HGG-MYCN cancer cell line to 639 small-molecule compounds (including FDA-approved, phase I–IV, and experimental drugs). All compounds were dispensed using a D300e Digital printer (Tecan) in a 6–8-point serial dilution covering a concentration range from 0.0043 to 25 μM in 1536-well plates (Corning). After 72 h treatment, cell viability was detected by CellTiter-Glo Luminescent cell viability assay (Promega) using a Spark plate reader (Tecan). Dose-response to compounds was measured based on a normalized area under the curve (AUC). The AUC values for all compounds can be found in Supplementary data 1. AUC data of additional glioma cell lines as well as healthy control fibroblasts were generously provided for comparison by Marc Remke and Nan Qin.

## Histology, immunohistochemistry, and FISH-analysis

For hematoxylin and eosin (H&E) and immunohistochemistry (IHC) stains, brain tissue was fixed in 4% paraformaldehyde/PBS for at least 12 h. The tissue was dehydrated, embedded in paraffin, and sectioned at 4 μm according to standard protocols. All IHC stains were performed on a Ventana System (Roche) using standard protocols. The following antibodies were employed: MYCN (Cell Signaling, 517053, 1:1000), P53 (Dako, M7001, 1:800), Ki67 (Abcam, ab16667, 1:100), Nestin (Abcam, ab221660, 1:2000), SOX2 (Abcam, ab97959, 1:200), OLIG2 (Millipore, AB9610, 1:200), Cre (Covance / DCS-diagnostics, PRB-106P, 1:100) and GFAP (Dako, M0761, 1:200). FISH analyses were performed to detect possible amplifications at the *MYCN* locus using standard procedures and a SPEC MYCN/2q11 dual color probe (Zytovision, Germany).

## RNA Sequencing (RNA Seq)

We employed 11 mouse HGG-MYCN tumor samples (6 male and 5 female) as well as 6 mouse SHH-MBs (sex not determined) and three control samples (2 male and 1 female) of the cerebellum and three of the olfactory bulb (2 male and 1 female). After isolation of total RNA with TRIzol (Invitrogen), RNA integrity was analyzed with the RNA 6000 Nano Chip on an Agilent 2100 Bioanalyzer (Agilent Technologies). From total RNA, mRNA was extracted using the NEBNext Poly(A) mRNA Magnetic Isolation module (New England Biolabs) and RNA-Seq libraries were generated using the NEXTFLEX Rapid Directional qRNA-Seq Kit (Bioo Scientific) as per the manufacturer´s recommendations. Concentrations of all samples were measured with a Qubit 2.0 Fluorometer (Thermo Fisher Scientific), and distribution of fragment lengths of the final libraries was analyzed with the DNA High Sensitivity Chip on an Agilent 2100 Bioanalyzer (Agilent Technologies). All samples were normalized to 2 nM and pooled equimolarly. The library pool was sequenced on the NextSeq500 (Illumina) with $1 \times 75$ bp read length and 16.1 to 18.6 Mio reads per sample.

For each sample, sufficient quality of the raw reads was confirmed by *FastQC* v0.11.8[49]. Afterwards, the reads were aligned to the mouse reference genome GRCm38 with *STAR* v2.6.1c[50] and simultaneously counted per gene by employing the *–quantmode GeneCounts* option. Counts are based on the Ensembl annotation release 95. Differentially expressed genes were estimated with *DESeq2* v1.22.2[51].

## RNA Sequencing analysis

Raw fastq files of mouse samples were processed in usegalaxy.eu[52]. Low quality reads were detected using FastQC (Galaxy Version 0.73+galaxy0). Trimmomatic (Galaxy Version 0.38.1) was used for trimming poor quality reads (reads with average quality <20). Reads were aligned to the mm39.ncbiRefseq.gtf.gz using STAR aligner (Galaxy Version 2.7.8a+galaxy1) and gene expression was quantified using featureCounts (Galaxy Version 2.0.1+galaxy2). Deseq2 (Galaxy Version 2.11.40.7+galaxy2) was used for generating VST-normalized

files for all samples. Human gene expression data was obtained from GSE73038[1]. Mouse samples were measured in two different batches, and the VST-normalized files were combined and corrected for batch effect using ComBat from sva package (3.44.0) in Rstudio (4.2.1). For comparing mouse (11 tumors with MYCN amp and *Trp53* mutation and 6 SHH tumors) and human data (173 samples), only orthologous genes were used ($n = 14,416$). In order to correct for species-specific batch effect, ComBat was applied after combining the mouse and human data (GSE73038[1]). Average tumor subgroup-specific gene expression was used for calculating euclidean distance. Sample-sample distance plot was visualized using ComplexHeatmap (2.12.1). Limma (3.52.2) was used for performing differential expression analysis. Top 500 differentially expressed genes (adjusted for multiple testing using Bonferroni-Hochberg correction and sorted by F-statistic) were visualized using umap (0.2.9.0) and ComplexHeatmap (2.12.1) in Rstudio. AGDEX was performed in C++ as described previously[53]. AGDEX was based on 14,416 orthologues genes with human sonic hedgehog medulloblastoma (SHH MB) and mouse SHH MB as reference group for differential expression of tumor samples in each species, respectively. The statistical significance of the observed AGDEX value was determined via permutation as described[39,53]. The significance of changes in expression of MYCN was determined by Welch's t-test of non-MYCN and HGG-MYCN human tumors and mouse OB vs. mouse HGG-MYCN. Gene set enrichment analysis (GSEA) was performed using the software GSEA v4.2.0 of the Broad Institute[54,55]. The gene set comprised 344 genes with transcription factor binding evidence in the MYCN-21190229-SHEP-21N-HUMAN profile from the CHEA Transcription Factor Binding Site Profiles dataset[56,57]. Genes were ranked by signal-to-noise ratio and statistical significance was determined via 1000 gene set permutations. $P < 0.05$ was considered significant.

## Global DNA-methylation analysis

Three mouse HGG-MYCN tumors were employed to methylation array (two male and one female). After isolation of genomic DNA, 200–500 ng DNA was used for bisulfite conversion by the EZ DNA Methylation Kit (Zymo Research). Then, the Mouse Methylation Bead Chip Array (Illumina) covering 285,000 CpG sites on the mouse genome was used on an iScan device (Illumina).

In order to compare the global DNA-methylation profile of mouse tumors with that of human samples, we collected human methylation profiles generated with the Illumina EPIC array, consisting of 850,000 CpG sites. The use of biopsy-specimens for research upon anonymization was always in accordance with local ethical standards and regulations at the University Medical Center Hamburg-Eppendorf. Apart from data generated for diagnostic purposes in-house and by cooperation partners, we included data generated and published by Capper et al. in 2018 (GSE73801)[42] for the overview of human brain tumor entities. The raw data were preprocessed and beta values were extracted with minfi[58]. The 10,000 most differentially methylated CpG sites were used to generate a UMAP with the UMAP package[59].

For the comparison of mouse and human tumors, only EPIC data were employed and $n = 431$ human brain tumors were included. Beta values were extracted from human idat files using minfi and the entire human dataset was quantile normalized. Mouse data were processed with SeSAMe[60] (v1.12.9), beta values were extracted and the data were quantile normalized. 141 identical CpG sites were extracted from both datasets and human and mouse datasets were combined. After combination, the mixed mouse-human dataset was quantile normalized and an UMAP was generated.

The heatmap of the copy number variation (CNV) of 47 HGG-MYCN was created from idat files. CNV data was generated with the Conumee package (v1.28.0)[61]. CNV values of the bin signals were calculated and depicted as a heatmap using ComplexHeatmap. Mouse CNV plots were created similarly by employing the conumee pipeline

with a few modifications including using an in-house generated reference set of normal mouse tissue.

## Single-cell RNA sequencing analysis

Mouse tumors of seven *hGFAP-cre::Trp53^{Fl/Fl}::lsl-MYCN^{Fl/wt}* mice (43–92 days old, 5 male and 2 females) were isolated and minced. Samples were digested for 30 min at 37 °C in a freshly prepared solution of papain (Worthington) in pre-warmed DMEM/F12 medium (Thermo Fisher) with DNase (Worthington) and passed through a 40 μm cell strainer (Corning). Red blood cells were depleted using ACK lysis buffer (Thermo Fisher), and single-cell suspensions were cryopreserved. After thawing, non-vital cells that stained positive for 7-Aminoactinomycin D (eBioscience) were removed using a BD FACS Aria II cell sorter (BD Biosciences). Approximately 10,000 vital cells were used as input for scRNA-seq. Single-index libraries were generated with Chromium Single Cell 3′ v3.1 technology (10x Genomics) and sequenced using a NextSeq 2000 sequencing instrument (high-throughput kit, 100 cycles) at the Genomics Core Facility (University Hospital Münster, Germany) after quality control using a Tapestation 2200 (Agilent Technologies). The samples were analyzed with the 10x Genomics CellRanger v6.0.2 pipeline[62] and Seurat R package v4.0.5[63]. Raw data were converted to fastq format with the CellRanger mkfastq function and then aligned against the murine reference transcriptome mm10 v2020-A with CellRanger count and default values. Seurat objects were generated for the samples based on the following filter criteria: at least three cells, a minimum feature count of 200, and cells with <25% of mitochondrial genes. Outlier cells with a high nCount_RNA value were classified as doublets and removed (threshold: 40,000–60,000). The filtered data were then normalized, integrated, and clustered with Seurat, using a resolution parameter of 0.5. Feature plots, UMAPs, dotplots, and heatmap visualizations were created with Seurat functions; a cluster-based cell type annotation was conducted based on the expression of characteristic marker genes per cell type. Lists of differentially expressed genes per cluster were calculated with Seurat's findMarkers function, using the MAST test "hinter" using a resolution parameter of 0.5. The genes for each signature depicted in Fig. 4 are shown in Supplementary Fig. 4.

Finally, a logistic regression analysis based on the approach of Young et al.[64] was used to identify similarities between the murine scRNA-seq clusters and a reference dataset by Mizrak et al.[65] (GSE134918). Briefly, a multinomial regression model was trained on the chosen reference dataset with the R package glmnet[66], and cell types were predicted for the original mouse data based on this model. Probabilities per reference cell type were aggregated on cluster level using R's mean function, and visualized as heatmaps with the package pheatmap[67].

## Statistical information

Animal survival was depicted as a Kaplan–Meier curve, made and assessed using GraphPad Prism 9 software. Significant *p* values are reported for appropriate figures in the figure legend. The sample size (*n*) is given in the figure legend or in the respective figure panel. Error bars represent the standard error of mean. The cut-off values for bioinformatic analyses are noted in the respective methods section. Further information is included in the Reporting summary.

## Reporting summary

Further information on research design is available in the Nature Portfolio Reporting Summary linked to this article.

## Data availability

The DNA methylation and RNA sequencing data including the raw data generated in this study have been deposited in the GEO database under accession code GSE227413. The scRNA sequencing data has been deposited in the GEO database under accession code GSE237237. The human gene expression data was obtained from GSE73038. Human DNA methylation data was obtained from GSE73801 and GSE215240. Mouse scRNA sequencing data publicly available from GSE134918 was employed in this study. AUC data from other glioma cell lines were provided by Marc Remke and Nan Qin. Source data are provided with this paper.

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

## Acknowledgements

We are thankful for the excellent technical support by Jacqueline Tischendorf, Celina Soltwedel, and Daniela Jeising. We like to thank the animal facility of the LIV Hamburg, Daniela Indenbirken, and the sequencing facility at the LIV Hamburg and the Fred Hutchinson Cancer Research Center for providing us with the pbt-04 cell line. A.K.A. and A.E. are thankful for the support within the interdisciplinary graduate school "Innovative Technologies in Cancer Diagnostics and Therapy" funded by the City of Hamburg. US received support from the Wilhelm-Sander Stiftung and from the Fördergemeinschaft Kinderkrebszentrum Hamburg. K.K. received support by the Kinderkrebshilfe Münster e. V. A.B. (Annika Ballast) was funded by the Medizinerkolleg (MedK) Muenster.

## Author contributions

Conceptualization, M.S. (Melanie Schoof), U.S., T.K.A., J.E.N., M.R. and K.K.; Performed experiments, M.S., S.G., T.K.A., M.D., C.W., A.B., N.Q., M.B.O., C.G., S.N., D.H., C.K., L.P., G.D.E., V.T., F.M., A.A., A.E., D.M. and M.Sp. (Michael Spohn), A.V.; Resources, M.B., M.M., S.R., Ma.S., J.V., N.S., D.T.W.J., M.R., J.E.N., K.K. and U.S.; Writing, M.S., T.K.A., U.S. and K.K.; Funding acquisition, U.S. and K.K. All authors provided critical feedback, helped shape the research and manuscript, and commented on the manuscript.

## Funding

## Competing interests

F.M. received support for meeting attendance from Servier, AbbVie, Incyte, Gilead, Jazz Pharmaceuticals, Novartis, Teva, Pfizer and Amgen; received support for medical writing from Servier and Springer Verlag; received research grants from Apis Technologies, Daiichi Sankyo and received speaker honoraria from Servier, Jazz Pharmaceuticals, AbbVie, Astellas Pharma, Bristol-Myers Squibb, MSD Sharp & Dohme, Novartis, Pfizer, and Stemline Therapeutics. The remaining authors declare no competing interests.

## Additional information

[1]Research Institute Children's Cancer Center, Hamburg, Germany. [2]Department of Pediatric Hematology and Oncology, University Medical Center, Hamburg-Eppendorf, Hamburg, Germany. [3]Center for Molecular Neurobiology (ZMNH), University Medical Center, Hamburg-Eppendorf, Hamburg, Germany. [4]Pediatric Hematology and Oncology, University Children's Hospital Muenster, Muenster, Germany. [5]Institute of Neuropathology, University Medical Center, Hamburg-Eppendorf, Hamburg, Germany. [6]Institute of Medical Informatics, University of Muenster, Muenster, Germany. [7]German Cancer Consortium (DKTK), Partner Site Essen/Düsseldorf, Düsseldorf, Germany. [8]Department of Pediatric Oncology, Hematology, and Clinical Immunology, Medical Faculty, Heinrich Heine University, University Hospital Düsseldorf, Düsseldorf, Germany. [9]Institute of Neuropathology, Heinrich Heine University, University Hospital Düsseldorf, Düsseldorf, Germany. [10]High-Throughput Drug Screening Core Facility, Medical Faculty, Heinrich-Heine-University Düsseldorf, Düsseldorf, Germany. [11]Mildred Scheel Cancer Career Center HaTriCS4 University Medical Center Hamburg-Eppendorf, Hamburg, Germany. [12]Hopp Children's Cancer Center (KiTZ), Heidelberg, Germany. [13]Pediatric Glioma Research Group, German Cancer Research Center (DKFZ), Heidelberg, Germany. [14]Department of Oncology, Hematology and Bone marrow transplantation, University Medical Center Hamburg-Eppendorf, Hamburg, Germany. [15]Division of Pediatric Neurooncology, German Cancer Consortium (DKTK), German Cancer Research Center (DKFZ), Heidelberg, Germany. [16]Department of Radiotherapy & Radiation Oncology, Hubertus Wald Tumorzentrum-University Cancer Center Hamburg (UCCH), University Medical Center Hamburg-Eppendorf, Hamburg, Germany. [17]These authors contributed equally: Kornelius Kerl, and Ulrich Schüller. ✉e-mail: u.schueller@uke.de

