## [Peer Review File · Nature Communications]

REVIEWER COMMENTS

Reviewer #1 (Remarks to the Author): Expert in high-grade glioma genomics and clinical research

Given the recent advances in pediatric glioma diagnosis and characterization of underlying genomic drivers, the authors of this manuscript generate and characterize a novel murine model for MYCN driven high grade glioma. They should be commended for their detailed characterization and clear description of the model, as well as attempts to show its value for clinical translation. The strength of this work is the characterization of a novel genetically engineered mouse model (GEMM) to aid treatment development and biological understanding of HGG driven by MYCN amplification. It is beautifully and clearly written. This model has potential to aid the field in understanding and developing treatment for MYCN-HGG. There are a few considerations that might make the work more impactful.

When characterizing the similarity of the HGG-MYCN GEMM to human tumors, the comparisons leave a few holes. In Figure 3c, the authors look at similarity of gene expression profiles between the HGG-MYCN GEMM and several human tumor types. It would be helpful to show significance of these similarities quantitatively (maybe * significance or give values in a table) as it is difficult to differentiate that murine HGG-MYCN is dissimilar from AT/RT, IDH-WT HGG, DMG and DHG, among others. It's unclear what the asterisk in panels C & D indicates (is this a significant relationship and the only significant relationship amongst the comparisons?). Clarification in the figure legend would be helpful.

The single cell analysis evaluating the cell of origin for HGG-MYCN GEMM is interesting, please clarify from what timepoint following birth these tumors were harvested and analyzed. Given the consistent temporal evolution of these tumors, it could be interesting to evaluate a range of very early time-points near the beginning of tumorigenesis to validate the authors' observation that these tumors arise from NSCs rather than a more-differentiated neuronal or oligodendroglial progenitor cells.

The authors' attempt to demonstrate translatability through a drug screen is applaudable. It's encouraging to see the screen returns hits that are consistent with currently known efficacious treatments for pediatric HGG. The decision to use a human MYCN-HGG for comparison also enhances plausibility. Please provide more information about the way the screen was constructed, which compounds were included, and why. Was the screen comprised solely of oncology-related compounds? Unbiased or selected based on blood brain barrier penetration / prior activity in glioma? Per methods section, it appears a dose response curve was generated for all these compounds. Please share more of this rich data set with readers. One could include a table with IC50 curves, growth curves, etc in the supplemental methods or upload the entire data set to a storage location. Additionally, the authors make a point that their top 3 hits (doxorubicin, SN38, and etoposide) are less effective in adult GBM lines. Frankly the data in Figs 5F-H don't completely support that statement. I recommend sharing additional data or rephrasing these statements.

Lastly, the obvious next experiment is to treat the GEMM with doxorubicin, SN38 or Etoposide as proof of principle. This seems a straightforward experiment and would generate valuable in vivo data. While probably not necessary for this manuscript, since these are already validated treatments for pHGG, the authors should consider using one or more of these treatments as controls in future experiments of novel therapeutics.

Small points:

Figure 3 A/B – legend indicates there are 3 HGG_MYCN tumors, but there appear to be 15+ pink dots. Please clarify whether each dot is a sample or a tumor and how many samples per tumor were evaluated.

Temozolomide, while a mainstay of therapy for adult glioma is generally not used for pediatric glioma and should probably not be compared as the standard chemotherapy treatment for this type of glioma in the discussion.

Reviewer #2 (Remarks to the Author): Expert in high-grade glioma genomics, clinical research, and mouse models

In this manuscript, Schoof et al. create and investigate a genetically engineered mouse model of pediatric HGG-MYCN and discern its intratumoral heterogeneity; they subsequently investigate drug susceptibilities using mouse and human tumor cells. Overall, this is a well conducted characterization of an important new model. The methodology appears sound, and the conclusions are adequately supported by the data. Below are comments by manuscript section:

Introduction

- It would be helpful if the authors discussed why these tumors were originally found in the PNET group even though they are gliomas; obviously the PNET group is heterogeneous, but is there something histologically distinct about these tumors, since most HGG were not classified as PNETs?
- It would be helpful to clarify “extremely poor prognosis”; was there a long-term survival rate measured or something else besides the 14-month median overall survival, which is similar to other PHGG?
- What percentage of these tumors carry these MYCN and TP53 alterations? That would help to understand the fidelity of the model to these tumors.

- MYCN is more associated with group 4 medullo than group 3 (more MYC-driven); the authors should clarify this in the discussion in line 84 (the tumors from this model may resemble group 3 medullo, but this conflict should be mentioned).

- Line 98: should be "as well as"

Results

- For 1a, the authors should clarify whether all these were pediatric tumors or at least tumors with high frequency in pediatrics.

- In 1d-e, "no alteration" is marked as gray, whereas that section of the pie charts is white.

- It would be helpful to show the detail of the MYCN amplification in 1h.

- Line 118 should be Fig 1h-j

- In 2f, it should be mentioned that the alterations are somewhat heterogeneous and not present in all tumors, while others are present in individual tumors.

- It would be helpful to show whether there were markers of neuronal/embryonal tumors for which both mouse and human tumors were negative.

- Line 161: delete "a"

- The purpose and significance of the results in lines 157-164 should be further clarified.

- More clarification of the meaning of the results in 3c-d would be helpful, especially since it looks like the mouse SHH MB are similar distance from the human HGG-MYCN and SHH-MB, and there is no similar MB control for 3d.

- In 3f, could mouse MYCN expression also be compared to expression in the mouse SHH-MB tumors and/or other normal mouse tissue outside the olfactory bulb?

- Why were only 9 of the tumor cell clusters included in the larger areas?

- Was scRNA-seq possible on any of the human tumors? If not, this should be stated.

- Line 234: should be "originate"

- Several of the drug hits from the screen, while they are used in some pediatric CNS tumors, have poor BBB penetration and are thus of limited utility (doxorubicin, daunorubicin).

- It appears that the three drugs tested in 5f-h also have high effectiveness against other PHGG, and also adult GBM for doxorubicin.

Discussion

- More discussion on differences between areas of tumor initiation would be helpful aside from speculation on the importance of sensory input, especially since there were no human tumors in the occipital lobes, which receive most visual input.
- Line 315: should be “was first discovered”
- Line 327: don’t need the word “two”
- Line 329: should be “unstable”
- Temozolomide is not a standard treatment used in PHGG (radiation is the only standard treatment). Some discussion of how this model might be used in in vivo studies would be useful, since this is a likely next step with the model.

Methods

- Methods are well described overall
- More information on statistical methodology should be given, e.g. for comparisons in 3e-f.

Reviewer #3 (Remarks to the Author): Expert in glioma genomics, genetically-engineered mouse models, and drug screening

The paper by Schoof and co-authors describes a new model of pediatric high-grade gliomas of the subclass MYCN (HGG-MYCN) in mice engineered to lack p53 and overexpress MYCN in CNS cells targeted by the hGFAPCre line. They show high incidence of forebrain tumors, which they claim are a good model of the human tumors as they seem to share histological and molecular features with them. They then analyse these murine tumors with single-cell RNA sequencing and show intratumoral heterogeneity with neuronal and oligodendroglial lineage signatures. Finally, they perform a drug screening on both mouse and human tumor cells to predict potential response to drugs already in use in other paediatric brain tumor patients.

I am not convinced this is a good model of the HGG-MYCN human tumors.

1) If only 60% of the human HGG-MYCN analysed show MYCN amplification, why are the rest (40%) classified as HGG-MYCN tumors at all? Is there MYCN overexpression in these tumors too and what is the mechanism? This is a crucial point to justify the modelling strategy, ie MYCN overexpression at genetic level.

2) Only 36% of the human tumors analysed carry mutations in p53 and MYCN, does this mean there is a significant proportion of these tumors that overexpress MYCN but do not lack p53? If this is the case do mice overexpressing MYCN alone develop these tumors? Probably not, as the authors mention in the

introduction referring to the work of other researchers. Are there different subgroups of HGG-MYCN tumors in human then and are the authors modelling only one of these subgroups in mice?

3) The authors clearly show an origin of the tumors in the olfactory bulb in their mice. This does not recapitulate the location/origin of these tumors in human

4) Single cell transcriptome analysis supports an origin from multipotent stem or progenitor cells in the olfactory bulb in the mouse, however the presence of these cells in the olfactory bulb in human is controversial, probably not existent and again the human tumor does not originate here.

5) A DNA methylation classification based on 141 CpG sites is rudimentary at best and may lack the granularity to robustly discriminate between tumor entities. Similarly the value of the AGDEX algorithm to assess transcriptomic similarities between human and murine tumor is controversial. The authors use a SHH MB mouse model as an internal control, but this model was never validated as a faithful model of human SHH MB over and above using a similar approach as done here, as such it cannot be used as a control for the analytical strategy used.

Other points

I struggle to see the added value of carrying out the drug screening in human and mouse lines. What is there to gain, given that the drug treatment efficacy hasn't been validated in the mice in vivo, which could have justified the approach. However, restricting the drug choice to those active on both human and mouse cells could skew the conclusions, if the mouse model is not as faithful as the authors claim.

hGFAPCre lines have been shown to lead to recombination of floxed constructs in the hindbrain in vivo, the authors claim this is not the case in their hands (line 137/138), assessed how? And why is this the case?

Reviewer #4 (Remarks to the Author): Expert in glioma genomics and single-cell RNA-seq

In "Mouse modeling of pediatric HGG-MYCN reveals intratumoral heterogeneity including neuronal and oligodendroglial lineage signatures," Schoof and colleagues present their work on diffuse pediatric-type diffuse high-grade gliomas H3-wildtype and IDH-wildtype, which in humans include three subtypes pHGG RTK1, pHGG RTK2, and pHGG MYCN. Each subtype is enriched for a distinct set of features. The pHGG MYCN subtype which is the focus of this study is the most aggressive of the three subtypes and has been reported to grow with a biphasic pattern of spindle-shaped and epithelioid cells, has large cells with prominent nucleoli, and grows with a mix of diffuse infiltration and highly cellular circumscribed nodules. Despite being gliomas, pHGG MYCN often do not express glial markers but rather they express neuronal markers. They retain expression of H3K27-trimethyl. To better understand these rare and aggressive pediatric brain tumors which harbor MYCN amplifications and TP53 mutations, the authors generated a genetically engineered mouse model that developed tumors resembling HGG-MYCN in humans – the tumors developed in supratentorial locations (in the forebrain olfactory bulb region, with

no cases seen in infratentorial areas), single-cell RNA sequencing revealed a high degree of intratumoral heterogeneity with neuronal and oligodendroglial lineages (and lack of the astrocytic and mesenchymal lineages observed in adult high grade gliomas), and methylation and transcriptional profiling revealed substantial similarity with human HGG-MYCN. Drug screening identified FDA approved brain penetrant drugs already used in pediatric glioma patients as potential therapies including doxorubicin, irinotecan, and etoposide.

This is a rigorous, well conducted study that carefully describes and characterizes a new GEMM for study a rare pediatric brain tumor. The model will be of high interest to pediatric neuro-oncology researchers. It would help if the authors more carefully described in the discussion section any limitations of the GEMM and differences with human that they think are important to report for future users (a separate detailed paragraph would be very useful). For example, they do report and discuss that the location differs somewhat from human tumors, and that no infratentorial examples were observed compared with a small fraction occurring in that location in humans. It appears that the GEMM tumors are largely circumscribed, which resembles human HGG-MYCN. What is the frequency of circumscribed (sharp demarcation from brain tissue) v. infiltrative growth in the GEMM, what is the frequency of necrosis or vascular proliferation in the GEMM, is the GFAP staining that is seen by IHC in the GEMM more than what is seen in human tumors (where it is typically seen in only infrequent scattered cells; is Fig 2m and Fig 2t a Cre and GFAP double stain?), do the GEMM tumors retain H3K27me3 expression, etc. Similarly, can the authors describe how the recurrent aberrations detected in chr 7 (loss), 14 (loss), 16 (gain) compare with chromosomal changes identified in the human tumors. In the human data, were all aberrations in TP53 mutations or were any deletions similar to that used in the GEMM?

It is not fully clear what is the source of the 2,514 tumors including 47 HGG-MYCN that were used in Figure 1, and how they relate to the 5 in house cases that are mentioned. Will the GEMM be deposited in a repository?

Rebuttal letter

Reviewer #1 (Remarks to the Author): Expert in high-grade glioma genomics and clinical research

Given the recent advances in pediatric glioma diagnosis and characterization of underlying genomic drivers, the authors of this manuscript generate and characterize a novel murine model for MCYN driven high grade glioma. They should be commended for their detailed characterization and clear description of the model, as well as attempts to show its value for clinical translation. The strength of this work is the characterization of a novel genetically engineered mouse model (GEMM) to aid treatment development and biological understanding of HGG driven by MYCN amplification. It is beautifully and clearly written. This model has potential to aid the field in understanding and developing treatment for MYCN-HGG. There are a few considerations that might make the work more impactful.

When characterizing the similarity of the HGG-MYCN GEMM to human tumors, the comparisons leave a few holes. In Figure 3c, the authors look at similarity of gene expression profiles between the HGG-MYCN GEMM and several human tumor types. It would be helpful to show significance of these similarities quantitatively (maybe * significance or give values in a table) as it is difficult to differentiate that murine HGG-MYCN is dissimilar from AT/RT, IDH-WT HGG, DMG and DHG, among others.

We thank the reviewer for this comment and are sorry for the color code, which was not very clear. We added the values for all analysis to the figure in order to make it more clear to the reader.

It's unclear what the asterisk in panels C & D indicates (is this a significant relationship and the only significant relationship amongst the comparisons?). Clarification in the figure legend would be helpful.

We apologize for the insufficient figure legend. The asterisk marks the value, which indicates the highest similarity, which is the smallest in Euclidean distance and the smallest p-value. We added the respective statement to the figure legend to make this clearer.

The single cell analysis evaluating the cell of origin for HGG-MYCN GEMM is interesting, please clarify from what timepoint following birth these tumors were harvested and analyzed. The tumors for single cell analysis were harvested as mice got symptomatic. The animals were between 77 and 92 days old. We added this information to the respective methods section. During the revision we also included two additional timepoints: 1.) n = 2 for p43; and 2.) n = 2 for p70 (see below).

Given the consistent temporal evolution of these tumors, it could be interesting to evaluate a range of very early time-points near the beginning of tumorigenesis to validate the authors' observation that these tumors arise from NSCs rather than a more-differentiated neuronal or oligodendroglial progenitor cells.

We highly appreciated this comment of the reviewer and agree that this analysis is interesting and improves the manuscript. Therefore, we performed the suggested experiment and generated scRNA sequencing data of four more mouse samples. We used two 43-day old mice in order to analyze early tumor lesions and two 70-day old mice. We then compared the tumor cells as well as tumor microenvironment and used logistic regression to compare the mouse tumor cells to the SVZ/VZ niche of mice. These analysis reveal some new findings: First, the TME is reduced upon later tumor stages, probably due to sampling of a more pure

tumor sample if the tumor is generally larger. Second, tumor cells of all tumor ages are most similar to adult neural stem/transit amplifying cells which strengthens our hypothesis of a cellular origin in this niche. And third, in early tumor lesion, only the oligodendroglial subpopulation is detected, the neuronal population then appears throughout tumor development and at the latest stage, a new, likely further differentiated, tumor population appears.

We included the new sequencing data in the manuscript and completely rebuild Figure 4 showing these findings.

The authors' attempt to demonstrate translatability through a drug screen is applaudable. It's encouraging to see the screen returns hits that are consistent with currently known efficacious treatments for pediatric HGG. The decision to use a human MYCN-HGG for comparison also enhances plausibility. Please provide more information about the way the screen was constructed, which compounds were included, and why. Was the screen comprised solely of oncology-related compounds? Unbiased or selected based on blood brain barrier penetration / prior activity in glioma? Per methods section, it appears a dose response curve was generated for all these compounds. Please share more of this rich data set with readers. One could include a table with IC50 curves, growth curves, etc in the supplemental methods or upload the entire data set to a storage location. Additionally, the authors make a point that their top 3 hits (doxorubicin, SN38, and etoposide) are less effective in adult GBM lines. Frankly the data in Figs 5F-H don't completely support that statement. I recommend sharing additional data or rephrasing these statements.

We thank the reviewer for the encouraging statement concerning our drug screen. We included a new phrase concerning the selection of substances included in the screen. Additionally, we added the AUC values for all compounds in a new supplementary table. We rephrased the statement concerning the efficacy in adult GBM.

Lastly, the obvious next experiment is to treat the GEMM with doxorubicin, SN38 or Etoposide as proof of principle. This seems a straightforward experiment and would generate valuable in vivo data. While probably not necessary for this manuscript, since these are already validated treatments for pHGG, the authors should consider using one or more of these treatments as controls in future experiments of novel therapeutics.

We appreciate this helpful comment from the reviewer concerning future in vivo experiments. We surely are planning such experiments in the future using our mouse model as well as existing PDX models for HGG-MYCN for treatment experiments. In those we want to include doxorubicin, SN38 and Etoposide, likely along with new more targeted therapies. We included an outlook statement in the discussion of the manuscript.

Small points:

Figure 3 A/B – legend indicates there are 3 HGG_MCYN tumors, but there appear to be 15+ pink dots. Please clarify whether each dot is a sample or a tumor and how many samples per tumor were evaluated.

We apologize for the confusion. We used eleven mouse tumors for RNA sequencing which is included in the figure legend but was obviously not clear enough. We changed the wording accordingly to make it more obvious.

Temozolomide, while a mainstay of therapy for adult glioma is generally not used for pediatric glioma and should probably not be compared as the standard chemotherapy treatment for this type of glioma in the discussion.

We thank the reviewer for this comment. We relied on the treatment regimen in Germany, which is based on the HIT-HGG 2007 study. Preliminary results of this show a survival

benefit upon clinical use of Temozolomide compared to cisplatin based chemotherapy (von Bueren et al., HGG-16. Final analysis of the HIT-HGG-2007 trial (ISRCTN19852453): Significant survival benefit for pontine and non-pontine pediatric high-grade gliomas in comparison to previous HIT-GBM-C/-D trials., Neuro-Oncology, Volume 24, Issue Supplement_1, June 2022, Pages i63–i64). Anyways, we rephrased the respective paragraph in the discussion to account for different treatments in other countries.

Reviewer #2 (Remarks to the Author): Expert in high-grade glioma genomics, clinical research, and mouse models

In this manuscript, Schoof et al. create and investigate a genetically engineered mouse model of pediatric HGG-MYCN and discern its intratumoral heterogeneity; they subsequently investigate drug susceptibilities using mouse and human tumor cells. Overall, this is a well conducted characterization of an important new model. The methodology appears sound, and the conclusions are adequately supported by the data. Below are comments by manuscript section:

Introduction

- It would be helpful if the authors discussed why these tumors were originally found in the PNET group even though they are gliomas; obviously the PNET group is heterogeneous, but is there something histologically distinct about these tumors, since most HGG were not classified as PNETs?

We thank the reviewer for his comment. These tumors were histologically diagnosed as PNETs as they present with undifferentiated, densely packed cell nuclei and highly circumscribed tumors without typical glial features. We added this information in the introduction.

- It would be helpful to clarify “extremely poor prognosis”; was there a long-term survival rate measured or something else besides the 14-month median overall survival, which is similar to other PHGG?

We appreciate the comment of the reviewer and clarified this statement. HGG-MYCN have recently been included in a large cohort of pediatric HGGs and displayed the worst prognosis (Sturm, D., Capper, D., Andreuolo, F. et al. Multiomic neuropathology improves diagnostic accuracy in pediatric neuro-oncology. Nat Med 29, 917–926 (2023)). We added this information in the introduction.

- What percentage of these tumors carry these MYCN and TP53 alterations? That would help to understand the fidelity of the model to these tumors.

We apologize for the lack of clarity regarding this aspect. As stated in figure 1, we identified genetic *MYCN* amplifications in 60% of cases and *TP53* mutations in 68%. Together these alterations occur in 36% of examined tumors.

- MYCN is more associated with group 4 medullo then group 3 (more MYC-driven); the authors should clarify this in the discussion in line 84 (the tumors from this model may resemble group 3 medullo, but this conflict should be mentioned).

We thank the reviewer for this comment and added this information to the introduction.

- Line 98: should be “as well as”

We apologize and changed the manuscript accordingly.

Results

- For 1a, the authors should clarify whether all these were pediatric tumors or at least tumors with high frequency in pediatrics.

We added the information that we included the most common brain tumors as well as potential differential diagnoses in the figure legend.

- In 1d-e, “no alteration” is marked as gray, whereas that section of the pie charts is white.

We apologize for the inconsistency and changed the figure accordingly.

- It would be helpful to show the detail of the MYCN amplification in 1h.

We appreciate this valuable hint of the reviewer and added a new figure panel to figure 1 including a representative CNV plot of a HGG-MYCN and a magnification of chromosome 2 including the MYCN amplification.

- Line 118 should be Fig 1h-j

We apologize and changed the manuscript accordingly.

- In 2f, it should be mentioned that the alterations are somewhat heterogeneous and not present in all tumors, while others are present in individual tumors.

We appreciate the reviewer`s comment and added a statement about the privacy of some alterations and the reoccurrence of others in the respective section.

- It would be helpful to show whether there were markers of neuronal/embryonal tumors for which both mouse and human tumors were negative.

We value the input of the reviewer and added a supplementary figure 2 showing that neither human nor mouse tumors express synaptophysin, NeuN or Neurofilament. In contrast, TubB3 was expressed in tumors of both species.

- Line 161: delete “a”

We apologize and changed the manuscript accordingly.

- The purpose and significance of the results in lines 157-164 should be further clarified.

We thank the reviewer for this comment and added more information on this aspect to the respective section. We decided to breed additional mouse models as the cell-of-origin as well as time and place of tumor onset for HGG-MYCN is unknown. Therefore, we decided to investigate tumor development in other cell populations of the developing CNS by breeding animals with the same genetic alterations in other target cell populations. We also included a statement concerning the meaning of embryonic lethality in our model in the discussion.

- More clarification of the meaning of the results in 3c-d would be helpful, especially since it looks like the mouse SHH MB are similar distance from the human HGG-MYCN and SHH-MB, and there is no similar MB control for 3d.

We thank the reviewer for this important comment. We added the values of the Euclidean distance as well as the AGDEX analysis in the figure to make this clearer.

- In 3f, could mouse MYCN expression also be compared to expression in the mouse SHH-MB tumors and/or other normal mouse tissue outside the olfactory bulb?

We value the suggestion of the reviewer and added the MYCN expression in healthy mouse cerebellum and the mouse SHH-MB tumors in the figure. Our HGG-MYCN tumors express comparable amounts of MYCN to the SHH-MB mouse tumors which are known to have a hyperactive SHH signaling leading to high MYCN expression.

- Why were only 9 of the tumor cell clusters included in the larger areas?

We sincerely apologize for not being able to answer to this question. Although trying hard, we did not understand exactly which 9 clusters you were talking about and what was meant by larger areas. We will certainly be very happy to answer your question once we have fully understood it. If, after seeing the revised version of the manuscript and the figures, your question is still relevant, please be kind enough to rephrase it so that we can understand more precisely what you mean. Thank you very much in advance!

- Was scRNA-seq possible on any of the human tumors? If not, this should be stated. We fully agree with the reviewer that scRNA-seq data of human HGG-MYCN would be interesting but we could not generate such data ourselves and so far, no single cell data of these rare tumors is available. We included this information in the manuscript.

- Line 234: should be “originate”
We apologize and changed the manuscript accordingly.

- Several of the drug hits from the screen, while they are used in some pediatric CNS tumors, have poor BBB penetration and are thus of limited utility (doxorubicin, daunorubicin). We fully agree with the reviewer that BBB penetration is of high importance for the utility of drugs in HGG-MYCN. Anyway, we included also drugs without proven BBB penetration as top hits as it has been shown before that the BBB is often altered in patients with brain tumors. Additionally, the selection doesn't limit to the blood-brain barrier permeability since, due to the recent development of materials science and nanotechnology, multiple strategies could be used to deliver drugs across BBB.

- It appears that the three drugs tested in 5f-h also have high effectiveness against other PHGG, and also adult GBM for doxorubicin. We thank the reviewer for his attentive consideration of our figures and agree that Etoposide, Doxorubicin and SN38 are also effective in killing other pedHGG and also some adult GBM. As none of the three substances are target specific for HGG-MYCN, this is not surprising. We decided in this first attempt to focus on drugs which could be easily translated into the clinic and would still improve patient survival. As a next step in the research on this new and rather undescribed tumor entity, finding targeted therapies for HGG-MYCN will be the focus in the future.

Discussion

- More discussion on differences between areas of tumor initiation would be helpful aside from speculation on the importance of sensory input, especially since there were no human tumors in the occipital lobes, which receive most visual input.

We appreciate the comment of the reviewer. We hypothesize that the tumor initiation is dependent on adult neurogenesis. The areas of adult neurogenesis clearly differ between mouse and humans. In mice, most adult neurogenesis occurs in the subventricular zone and cells migrate into the olfactory bulb which is exactly the place of tumor initiation in ours and other mouse brain tumor models. In human, most adult neurogenesis is strongest in the striatum (as reviewed for example here: Ernst A, Frisén J (2015) Adult Neurogenesis in Humans- Common and Unique Traits in Mammals. PLOS Biology 13(1): e1002045.). This difference together with sensory input might explain the difference in tumor location. Clearly, this hypothesis needs further proof in the future but we added this in to discussion section.

- Line 315: should be “was first discovered”
We apologize and changed the manuscript accordingly.

- Line 327: don't need the word "two"

We apologize and changed the manuscript accordingly.

- Line 329: should be "unstable"

We apologize and changed the manuscript accordingly.

- Temozolomide is not a standard treatment used in PHGG (radiation is the only standard treatment). Some discussion of how this model might be used in *in vivo* studies would be useful, since this is a likely next step with the model.

We thank the reviewer for this comment. We relied on the treatment regimen in Germany, which is based on the HIT-HGG 2007 study. Preliminary results of this show a survival benefit upon clinical use of Temozolomide compared to cisplatin based chemotherapy (von Bueren et al., HGG-16. Final analysis of the HIT-HGG-2007 trial (ISRCTN19852453): Significant survival benefit for pontine and non-pontine pediatric high-grade gliomas in comparison to previous HIT-GBM-C/-D trials., *Neuro-Oncology*, Volume 24, Issue Supplement_1, June 2022, Pages i63–i64). Anyways, we rephrased the respective paragraph in the discussion to account for different treatments in other countries.

We included a short outlook on potential *in vivo* treatment studies which could include the comparison of the current treatment with radiation and Temozolomide and Irinotecan, Etoposide, Doxorubicin and any combination of those substances. Additionally, more targeted treatments described for other MYCN driven tumors like for example AURKA-, CDK2/9- and DNA replication repair inhibitors could be tested. Additionally, it might be useful to employ our mouse model as well as PDX models of HGG-MYCN in such treatment studies.

Methods

- Methods are well described overall

- More information on statistical methodology should be given, e.g. for comparisons in 3e-f.

We thank the reviewer and added the statistical information for the comparisons in the methods section.

Reviewer #3 (Remarks to the Author): Expert in glioma genomics, genetically-engineered mouse models, and drug screening

The paper by Schoof and co-authors describes a new model of pediatric high-grade gliomas of the subclass MYCN (HGG-MYCN) in mice engineered to lack p53 and overexpress MYCN in CNS cells targeted by the hGFAPCre line. They show high incidence of forebrain tumors, which they claim are a good model of the human tumors as they seem to share histological and molecular features with them. They then analyse these murine tumors with single-cell RNA sequencing and show intratumoral heterogeneity with neuronal and oligodendroglial lineage signatures. Finally, they perform a drug screening on both mouse and human tumor cells to predict potential response to drugs already in use in other paediatric brain tumor patients.

I am not convinced this is a good model of the HGG-MYCN human tumors.

1) If only 60% of the human HGG-MYCN analysed show MYCN amplification, why are the rest (40%) classified as HGG-MYCN tumors at all? Is there MYCN overexpression in these tumors too and what is the mechanism? This is a crucial point to justify the modelling strategy, ie MYCN overexpression at genetic level.

We thank the reviewer for his comments and clearly appreciate that it might be surprising at first glance that only 60% of human HGG-MYCN carry genetic *MYCN* alterations. However, this rather new tumor entity is defined by a distinct DNA methylation and transcriptomic

pattern which is shared by all tumors regardless of having *MYCN* amplifications or not. So far, no molecular drivers apart from *MYCN* and *TP53* have been described, and no alternative mechanism for tumors without *MYCN* amplification has been identified so that these tumors seem to phenocopy the *MYCN* amplification even without genetic amplification of the oncogene. So, since the majority of tumors is driven by genetic alterations of *MYCN* and alternative mechanisms are unknown by now, we thought that modeling the disease accordingly in a mouse is reasonable.

2) Only 36% of the human tumors analysed carry mutations in p53 and *MYCN*, does this mean there is a significant proportion of these tumors that overexpress *MYCN* but do not lack p53? If this is the case do mice overexpressing *MYCN* alone develop these tumors? Probably not, as the authors mention in the introduction referring to the work of other researchers. Are there different subgroups of HGG-*MYCN* tumors in human then and are the authors modelling only one of these subgroups in mice?

We acknowledge the comment of the reviewer and as stated for *MYCN*, also *TP53* mutations are only detected in a proportion of patients and the two alterations don't always co-occur. Anyway, as the reviewer already mentioned, we cited other researchers, which could show that the introduction of only the *MYCN* transgene does not lead to brain tumor development and *TP53* loss alone induced diffuse glioma formation significantly later in life. As for tumors with or without *MYCN* amplification, also tumors with or without genetic loss of *TP53* do not seem to differ in their epigenetic profile, at least as far as we know until now. Therefore, and due to the molecular and histological similarity of our mouse model to the human tumors, we are convinced that our mouse model is still a reliable model for the human disease.

3) The authors clearly show an origin of the tumors in the olfactory bulb in their mice. This does not recapitulate the location/origin of these tumors in human

We fully agree with the reviewer that the location of the tumors in mice does not reflect the human tumor locations, but we believe that this rather reflects species differences than differences between mouse and human HGG-*MYCN*. These species differences are obvious in the major sensory input which can support tumor formation as well as in the areas of adult neurogenesis in the murine and human CNS. We hypothesize that the tumor initiation is dependent on adult neurogenesis. The areas of adult neurogenesis clearly differ between mouse and humans. In mice, most adult neurogenesis occurs in the subventricular zone and cells migrate into the olfactory bulb which is exactly the place of tumor initiation in our and other mouse brain tumor models. In human, most adult neurogenesis is strongest in the striatum (as reviewed for example here: Ernst A, Frisén J (2015) Adult Neurogenesis in Humans- Common and Unique Traits in Mammals. *PLOS Biology* 13(1): e1002045.). This difference together with sensory input might explain the difference in tumor location. Clearly, this hypothesis needs further proof in the future, and we added this to discussion section.

4) Single cell transcriptome analysis supports an origin from multipotent stem or progenitor cells in the olfactory bulb in the mouse, however the presence of these cells in the olfactory bulb in human is controversial, probably not existent and again the human tumor does not originate here.

We agree with the reviewer that the existence of progenitor cells in the human olfactory bulb is controversial. Anyway, as discussed before, adult neurogenesis is regionally distinct in the murine and human brain. Therefore, it is not completely surprising that the tumors appear in different brain regions in the mouse model than in their human counterpart. Despite these differences, the tumor initiating cell types are most likely similar, which is reflected by the high similarity in histology, DNA methylation, and gene expression.

5) A DNA methylation classification based on 141 CpG sites is rudimentary at best and may lack the granularity to robustly discriminate between tumor entities. Similarly the value of the AGDEX algorithm to assess transcriptomic similarities between human and murine tumor is controversial. The authors use a SHH MB mouse model as an internal control, but this model was never validated as a faithful model of human SHH MB over and above using a similar approach as done here, as such it cannot be used as a control for the analytical strategy used.

We agree with the reviewer that 141 CpG sites are rather few. However, UMAP analyses very nicely show that already these few sites are able to discriminate the different human brain tumor entities and might therefore also be able to discriminate different mouse tumor entities. We also agree that AGDEX analysis and Euclidean distance calculations all have their drawbacks. Still, if all kinds of different analyses (even if they may have drawbacks) result in a high similarity between murine and human tumors, this represents at least some evidence for having generated a good model. Anyway, we are open for further suggestions on how to compare the tumors best.

We used a SHH MB model as an internal control as this model is widely accepted and used for studying SHH MB. We and others think that this model recapitulates the human tumors convincingly as it is generated from the same cell-of-origin with the exact same genetic drivers as the human disease (see for example PMID: 18691547, PMID: 24871706).

Other points

I struggle to see the added value of carrying out the drug screening in human and mouse lines. What is there to gain, given that the drug treatment efficacy hasn't been validated in the mice *in vivo*, which could have justified the approach. However, restricting the drug choice to those active on both human and mouse cells could skew the conclusions, if the mouse model is not as faithful as the authors claim.

As explained above, we do indeed think that our model reliably models the human disease and that we will need the model und identify therapeutic target and suitable drugs. We therefore think that including the model in our experiments for findings cures is useful. Anyway, all top substances are also among the top hits if only the human cells are considered. And, of course, we will surely perform additional *in vivo* experiments validating the efficiency of our top candidates and included a short outlook on those experiments in the discussion.

hGFAPCre lines have been shown to lead to recombination of floxed constructs in the hindbrain *in vivo*, the authors claim this is not the case in their hands (line 137/138), assessed how? And why is this the case?

We apologize for the confusion of the reviewer and rephrased the sentence to make it clearer. We surely expect recombination of the floxed alleles in the hindbrain of our mice but we did not detect any additional MYCN expression or other histological features that could indicate tumor formation.

Reviewer #4 (Remarks to the Author): Expert in glioma genomics and single-cell RNA-seq

In "Mouse modeling of pediatric HGG-MYCN reveals intratumoral heterogeneity including neuronal and oligodendroglial lineage signatures," Schoof and colleagues present their work on diffuse pediatric-type diffuse high-grade gliomas H3-wildtype and IDH-wildtype, which in humans include three subtypes pHGG RTK1, pHGG RTK2, and pHGG MYCN. Each

subtype is enriched for a distinct set of features. The pHGG MYCN subtype which is the focus of this study is the most aggressive of the three subtypes and has been reported to grow with a biphasic pattern of spindle-shaped and epithelioid cells, has large cells with prominent nucleoli, and grows with a mix of diffuse infiltration and highly cellular circumscribed nodules. Despite being gliomas, pHGG MYCN often do not express glial markers but rather they express neuronal markers. They retain expression of H3K27-trimethyl. To better understand these rare and aggressive pediatric brain tumors which harbor MYCN amplifications and TP53 mutations, the authors generated a genetically engineered mouse model that developed tumors resembling HGG-MYCN in humans – the tumors developed in supratentorial locations (in the forebrain olfactory bulb region, with no cases seen in infratentorial areas), single-cell RNA sequencing revealed a high degree of intratumoral heterogeneity with neuronal and oligodendroglial lineages (and lack of the astrocytic and mesenchymal lineages observed in adult high grade gliomas), and methylation and transcriptional profiling revealed substantial similarity with human HGG-MYCN. Drug screening identified FDA approved brain penetrant drugs already used in pediatric glioma patients as potential therapies including doxorubicin, irinotecan, and etoposide.

This is a rigorous, well conducted study that carefully describes and characterizes a new GEMM for study a rare pediatric brain tumor. The model will be of high interest to pediatric neuro-oncology researchers. It would help if the authors more carefully described in the discussion section any limitations of the GEMM and differences with human that they think are important to report for future users (a separate detailed paragraph would be very useful). For example, they do report and discuss that the location differs somewhat from human tumors, and that no infratentorial examples were observed compared with a small fraction occurring in that location in humans. It appears that the GEMM tumors are largely circumscribed, which resembles human HGG-MYCN. What is the frequency of circumscribed (sharp demarcation from brain tissue) v. infiltrative growth in the GEMM, what is the frequency of necrosis or vascular proliferation in the GEMM, is the GFAP staining that is seen by IHC in the GEMM more than what is seen in human tumors (where it is typically seen in only infrequent scattered cells; is Fig 2m and Fig 2t a Cre and GFAP double stain?), do the GEMM tumors retain H3K27me3 expression, etc. Similarly, can the authors describe how the recurrent aberrations detected in chr 7 (loss), 14 (loss), 16 (gain) compare with chromosomal changes identified in the human tumors. In the human data, were all aberrations in TP53 mutations or were any deletions similar to that used in the GEMM? We highly appreciate the feedback of the reviewer. We clearly acknowledge the idea of a separate paragraph in the discussion, which states the limitations of our model and included this in the revised manuscript.

In order to account for a more detailed histologic characterization of our mouse tumors which enables comparison with human data, we added a new supplementary figure with detailed pictures of mouse HGG-MYCN histology and additional histological markers. We also performed H3K27me3 staining of mouse and human tumors, but the staining was technically not convincing in the mouse tissue and we therefore excluded this from the manuscript. The mouse tumors present with a relatively sharp demarcation from brain tissue. In some tumors, we observed cellular clusters infiltrating the brain parenchyma. Frequent histologic features of the tumors include giant cells and necrosis. Vascular proliferation was not detected in any of the tumors. We have not performed an in depth comparison to human tumor histology as detailed and large scale studies on HGG-MYCN histology are still missing.

We tried to compare the recurrent chromosomal aberrations that we observe in our mouse model to the chromosomal aberrations found in human tumors, but could not recapitulate the exact changes although the overall genetic instability is conserved between species. For a definite comparison of human and mouse tumors, more data on human as well as on mouse

tumors and the overall comparability of CNV changes in mouse models to human tumors is needed.

Regarding the P53 aberrations, we don't have data for all tumors, but most of the alterations were point mutations, which are associated with a loss of function of the protein. We believe that therefore the effect on the cells is comparable to the broad deletion in our GEMM.

It is not fully clear what is the source of the 2,514 tumors including 47 HGG-MYCN that were used in Figure 1, and how they relate to the 5 in house cases that are mentioned. Will the GEMM be deposited in a repository?

We apologize for the lack of clarity on the source of the data. We collected data from different sources including the Heidelberg Classifier reference set and in-house data as well as external data generated for diagnostic purposes. The 47 HGG-MYCN are derived from the reference set and other published resources and includes the five in-house cases. We clarified that in the methods section. We are not planning to deposit our GEMM in a repository as we did not engineer completely new transgenes but instead combined already described transgenic alleles, which are already available in repositories or from the respective scientists. Information on the original publications and alleles can be found in the methods section.

REVIEWER COMMENTS

Reviewer #1 (Remarks to the Author):

The authors have attempted to address all my and the other reviewers questions and have made several changes that improve the clarity and scientific interest of the manuscript. A few, relatively minor points, remain to be clarified.

For data rigor and reproducibility the authors should have at least 1 more mouse at the P92 timepoint (currently n=1), particularly since most cells belong to a “new and unique cell population” (lines 292-293).

Figure 5: The authors state in the legend: “top three substances Doxorubicin, SN-38 (active metabolite of Irinotecan) and Etoposide are efficiently impairing growth of HGG-MYCN cells, while not being as effective in adult GBM lines”. I would tone this down to match the updated language in the text, which is more reflective of the data (line 313).

Reviewer #2 (Remarks to the Author):

The authors have done well in addressing the critiques; I figured out my own answer to the question about the scRNA-seq that was not clear to the authors, so nothing further needed for this question. I have no further suggested revisions. I continue to feel that this is an important paper with sound methods that is scientifically and clinically relevant, and that the conclusions are supported by the data.

Reviewer #3 (Remarks to the Author):

I continue to have reservations about the translational relevance of this model for human neuro-oncology. The authors have addressed some of my concerns in their rebuttal however, no additional experiments have been performed to convince this reviewer of the validity of the translational soundness of this model. The authors reiterate they believe it does, I wish they would have attempted to convince me with additional experimental evidence.

Reviewer #4 (Remarks to the Author):

The authors have adequately address my questions and concerns with modifications to the text and with the new supplementary figures on histology.

REBUTTAL LETTER

Reviewer #1:

The authors have attempted to address all my and the other reviewers questions and have made several changes that improve the clarity and scientific interest of the manuscript. A few, relatively minor points, remain to be clarified.

For data rigor and reproducibility the authors should have at least 1 more mouse at the P92 timepoint (currently n=1), particularly since most cells belong to a “new and unique cell population” (lines 292-293).

We fully understand the comment of the reviewer that a conclusion based on one sample is not scientifically ideal and apologize for this. Unfortunately, there is no archived material or data from other mice of that age available. So, new mice would need to be generated. While this is no problem in general, the Kaplan-Meier survival curve in Figure 2b clearly shows that the vast majority of animals does not survive 92 days. In fact, the survival proportion on day 90 is only 4.4%. We would therefore need to breed dozens, if not more than a hundred mice in order to obtain a single mouse that carries the correct genotype and survives 92 days. We still agree that the respective data are scientifically useful, but we are unsure whether they are worth this enormous effort and animal production. Lastly, the experiment is certainly not realistic with the suggested timeframe given by the editor for this revision.

We also discussed about excluding this exceptional tumor from the analysis but decided to keep it in the manuscript for scientific clarity and to not withhold any data. We generated the data of this specific tumor, as we collected the cells as soon as mice got symptomatic, and we were not aware of the rarity of this long survival at that time. The exceptionally long survival might also be the reason for the differentness of the P92 tumor.

We are truly sorry for not being able to easily increase the number of samples of this tumor stage and are very cautious with drawing any conclusions. We further commented on this issue in the revised manuscript.

Figure 5: The authors state in the legend: “top three substances Doxorubicin, SN-38 (active metabolite of Irinotecan) and Etoposide are efficiently impairing growth of HGG-MYCN cells, while not being as effective in adult GBM lines”. I would tone this down to match the updated language in the text, which is more reflective of the data (line 313).

We thank the reviewer for this remark and changed the figure legend accordingly. We apologize for not changing this in the first revision.

Reviewer #2 (Remarks to the Author):

The authors have done well in addressing the critiques; I figured out my own answer to the question about the scRNA-seq that was not clear to the authors, so nothing further needed for this question. I have no further suggested revisions. I continue to feel that this is an important paper with sound methods that is scientifically and clinically relevant, and that the conclusions are supported by the data.

We thank the reviewer for his empowering words on our manuscript.

Reviewer #3 (Remarks to the Author):

I continue to have reservations about the translational relevance of this model for human neuro-oncology. The authors have addressed some of my concerns in their rebuttal however, no additional experiments have been performed to convince this reviewer of the validity of the translational soundness of this model. The authors reiterate they believe it does, I wish they would have attempted to convince me with additional experimental evidence.

We are sorry that we could still not convince the reviewer of the relevance of our mouse model. We further strengthened the limitations of our model in the discussion to make clearer that we are aware of those limitations. Anyways, as there is no current other model for this tumor entity and there is very limited human material, we still think that our mouse model adds more knowledge on HGG-MYCN.

Reviewer #4 (Remarks to the Author):

The authors have adequately address my questions and concerns with modifications to the text and with the new supplementary figures on histology.

We thank the reviewer for his positive feedback on our revised manuscript and are happy that we could answer all questions and concerns.

REVIEWERS' COMMENTS

Reviewer #1 (Remarks to the Author):

The authors have provided satisfactory answers or responses to all my questions.

Reviewer #3 (Remarks to the Author):

The authors have suitably addressed the limitations of their model in the discussion.